# CAPTAIN: Conformal-Prediction-Based Multi-Source Time-Series Forecasting

**Shuaicheng Zhang**[*]                                              *zshuai8@vt.edu*
*Virginia Tech*
*Blacksburg, VA, USA*

**Tuo Wang**[*]                                                     *tuowang@vt.edu*
*Virginia Tech*
*Blacksburg, VA, USA*

**Stephen Adams**                                              *scadams21@vt.edu*
*Virginia Tech National Security Institute*
*Arlington, VA, USA*

**Sanmitra Bhattacharya**                          *sanmbhattacharya@deloitte.com*
*Deloitte & Touche LLP*
*New York City, NY, USA*

**Sunil Reddy Tiyyagura**                              *sutiyyagura@deloitte.com*
*Deloitte & Touche Assurance & Enterprise Risk Services India Private Limited*
*Hyderabad, Telangana, India*

**Edward Bowen**                                              *edbowen@deloitte.com*
*Deloitte & Touche LLP*
*New York City, NY, USA*

**Balaji Veeramani**                                      *bveeramani@deloitte.com*
*Deloitte Consulting LLP*
*New York City, NY, USA*

**Dawei Zhou**                                                     *zhoud@vt.edu*
*Virginia Tech*
*Blacksburg, VA, USA*

**Reviewed on OpenReview:** *https://openreview.net/forum?id=WJjlXHo4yS*

[*]Equal contribution.

## Abstract

Uncertainty quantification is critical for real-world forecasting applications such as predictive maintenance, patient health monitoring, and environmental sensing, where decisions must account for confidence levels. Multi-source time-series forecasting introduces additional complexity due to inter-source interactions and temporal dependencies, which existing methods struggle to capture within a unified probabilistic framework. Most previous approaches also lack theoretical guarantees, leading to miscalibrated uncertainty estimates. We propose CAPTAIN: ConformAl Prediction based multi-source Time-series forecAstINg, a two-stage framework for uncertainty quantification in multi-source time-series forecasting. First, CAPTAIN employs Normal Inverse Gamma (NIG) distributions to model source-specific uncertainties and integrates a meta-source to capture inter-source interactions. Next, temporal copulas model the evolution of joint uncertainties over time, ensuring robust and

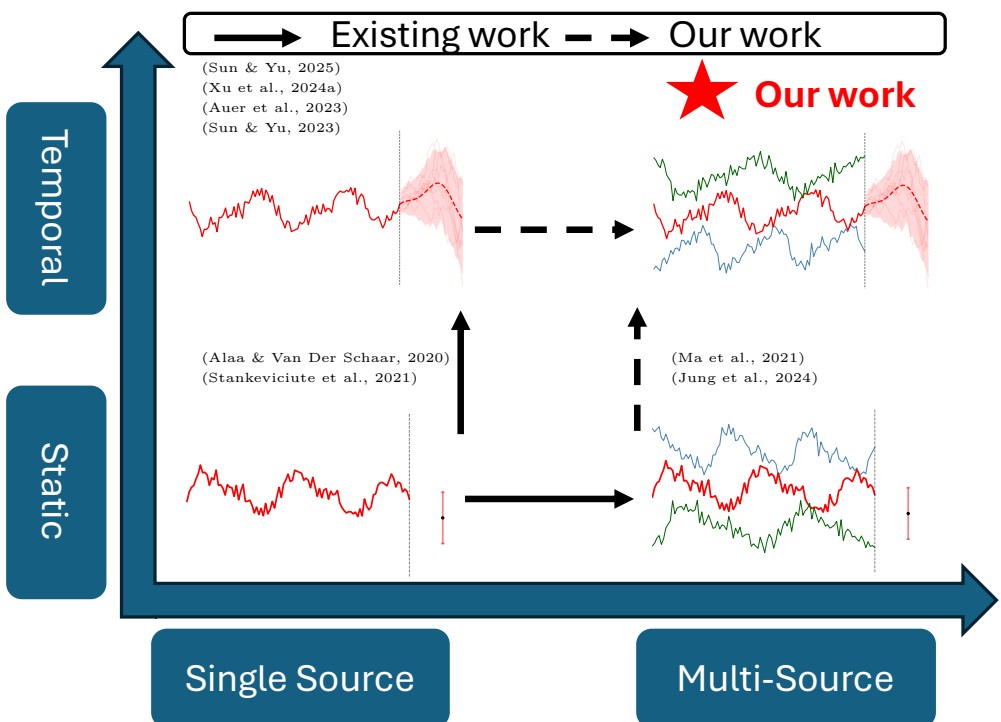

Figure 1: Positioning of CAPTAIN in multi-source time-series forecasting. Existing methods address either temporal dependencies (top-left) or multi-source fusion (bottom-right) independently. CAPTAIN uniquely addresses both dimensions simultaneously.

theoretically valid uncertainty coverage. Experiments on five diverse datasets (Synthetic, Shaoxing ECG, Air Quality, NGSIM Traffic, and ETTh1) demonstrate that CAPTAIN **achieves valid coverage ($\geq 90\%$) across all five benchmarks** while other baselines remain hard to achieve valid. Our empirical results on validity and width of coverage intervals, together with ablation studies, demonstrate CAPTAIN is a better approach for multi-source uncertainty quantification over existing state-of-the-art baselines.

# 1 Introduction

Multi-source time-series forecasting is fundamental to real-world applications including predictive maintenance in industrial systems (Sinha, 2022; Zonta et al., 2022), continuous patient health monitoring (AlSaad et al., 2024), and urban mobility analysis (Zhu et al., 2023). These applications integrate diverse data sources such as physiological signals (e.g., ECG, blood pressure), environmental measurements (e.g., temperature, air quality), and transportation data, each capturing complementary aspects of the underlying system. While prior work has advanced multi-source fusion for improved predictive accuracy (Baltrušaitis et al., 2018; Ramachandram & Taylor, 2017), the critical challenge of *uncertainty quantification* (UQ) remains largely unaddressed, limiting trustworthiness in safety-critical deployments.

In practice, data quality varies across sources and fluctuates over time, rendering point predictions insufficient for reliable decision-making (Liang et al., 2019; Lee et al., 2021; Ulmer & Cinà, 2021; Han et al., 2020). Consider patient monitoring: sensor misalignment or patient movement can degrade ECG signals while blood pressure readings remain reliable. Traditional multi-source models (Yang et al., 2017; Afyouni et al., 2022) assume uniform data quality across sources, potentially propagating corrupted signals and causing false alarms or missed critical events. A robust forecasting framework must therefore jointly model predictions and their associated uncertainties while dynamically adapting to source reliability.

Existing multi-source forecasting methods primarily target predictive accuracy through various fusion strategies such as combining raw features, merging learned representations, or aggregating source-specific pre-

dictions (Baltrušaitis et al., 2018; Ramachandram & Taylor, 2017). These approaches do not inherently quantify predictive uncertainty. Standard uncertainty estimation techniques can be applied post-hoc: MC Dropout (Gal & Ghahramani, 2016) approximates Bayesian inference through stochastic forward passes, while Deep Ensembles (Lakshminarayanan et al., 2017) aggregate predictions from independently trained networks. However, neither provides formal coverage guarantees, and prediction intervals may systematically under- or over-cover without explicit calibration (Guo et al., 2017; Kuleshov et al., 2018).

Conformal prediction (CP) offers a compelling alternative: a distribution-free framework providing finite-sample coverage guarantees without distributional assumptions (Vovk et al., 2005). While CP has been successfully extended to classification (Romano et al., 2020; Angelopoulos et al., 2020) and regression (Romano et al., 2019) settings, standard CP relies on *exchangeability*, an assumption violated in time-series data where temporal dependencies induce correlation across observations. Recent work addresses this limitation for single-source temporal forecasting (Stankeviciute et al., 2021; Sun & Yu, 2023; Xu et al., 2024a), yet these methods fail to capture *inter-source dependencies*: the correlations between uncertainty estimates across different data sources that are critical for multi-source fusion. This gap motivates two fundamental research questions:

- **Q1 (Inter-Source Temporal Uncertainty):** How can we develop a unified probabilistic framework that captures joint uncertainty patterns across *both* data sources and temporal dimensions?

- **Q2 (Theoretical Guarantees):** Can we design an uncertainty-aware framework that provides valid coverage guarantees while accommodating temporal dependencies intrinsic to multi-source forecasting?

We propose CAPTAIN (**C**onform**A**l **P**rediction based multi-source **T**ime-series forec**A**st**IN**g), a two-stage framework that addresses both questions through a principled integration of Normal Inverse Gamma (NIG) distributions and temporal copulas.

To address **Q1**, we employ NIG distributions to model source-specific uncertainties, leveraging their conjugate prior structure to jointly capture predictive mean and variance. The precision parameter $\gamma$ governs confidence weighting, enabling automatic down-weighting of unreliable sources during fusion. Building on (Ma et al., 2021), we derive a closed-form *NIG evidence fusion* operator that combines source-specific posteriors while preserving the NIG distributional family. Additionally, we introduce a learnable *meta-source* that captures inter-source dependencies not modeled by independent source assumptions.

To address **Q2**, we employ temporal copulas to model the joint evolution of non-conformity scores across time steps, inspired from (Sun & Yu, 2023). This integration yields prediction intervals with formal coverage guarantees that account for both temporal and inter-source dependencies.

In summary, our contributions are:

• **Problem Formulation:** We formalize uncertainty quantification for multi-source time-series forecasting, identifying the joint modeling of inter-source and temporal uncertainties as the key technical challenge (Section 2).

• **Algorithmic Framework:** We introduce CAPTAIN, combining (i) a Multi-Source NIG Encoder with learnable meta-source for structured uncertainty fusion, and (ii) a Copula-based Temporal Calibrator for coverage-guaranteed prediction intervals (Section 4).

• **Theoretical Analysis:** We provide mathematical proofs to show CAPTAIN maintains valid coverage under the temporal copula framework, establishing that NIG evidence fusion preserves the exchangeability structure required for conformal guarantees (Section 3).

• **Empirical Validation:** Experiments on five diverse datasets demonstrate that CAPTAIN **achieves valid coverage ($\geq 90\%$) across all five benchmarks** while other baselines achieve valid coverage on 4 or fewer benchmarks. On the challenging ETTh1 dataset, CAPTAIN achieves 92.8% coverage while CF-LSTM (88.4%) and MultiDimSPCI (87.6%) fail to reach the 90% target (Section 5). Code and data are publicly available.[1]

---

[1] https://github.com/zshuai8/2026-TMLR-CAPTAIN

**Related work.** Extended discussion of related work on multi-source forecasting, uncertainty quantification, and conformal prediction in time series is provided in Appendix A.10.

## 2  Preliminaries

We use boldface lowercase for vectors ($\mathbf{v}$) and boldface uppercase for matrices ($\mathbf{A}$). Let $\mathcal{Z} = \{z_1, \ldots, z_k\}$ denote $k$ data sources observed over horizon $T$. For source $z_l$ at time $t_j$: $\mathbf{x}_{z_l}^{t_j}$ denotes input features, $y_{z_l}^{t_j}$ the ground truth, $\hat{y}_{z_l}^{t_j}$ the prediction, and $\Gamma_{z_l}^{t_j}$ the prediction set. NIG parameters are $(\delta, \gamma, \alpha, \beta)$, and target coverage is $1 - \epsilon$.

### 2.1  Uncertainty Quantification via Conformal Prediction

Conformal prediction (CP) Vovk et al. (2005) provides distribution-free prediction sets with finite-sample coverage guarantees. Given calibration data and non-conformity scores, CP constructs sets guaranteed to contain true outcomes with probability $\geq 1 - \epsilon$. However, standard CP assumes exchangeability, which is violated in time-series data due to temporal dependencies and further complicated in multi-source settings. Our framework addresses this limitation. See Appendix A.2.2 for detailed CP background.

### 2.2  Problem Definition

We now formalize uncertainty-aware multi-source time-series forecasting. Unlike prior work focusing solely on predictive accuracy Afyouni et al. (2022); Yang et al. (2017), we require prediction sets with formal coverage guarantees despite temporal and inter-source dependencies.

**Problem 2.1** (Uncertainty-Aware Multi-Source Time-Series Forecasting). **Given:**

- Sources $\mathcal{Z} = \{z_1, \ldots, z_k\}$ observed over horizon $\mathcal{T} = \{t_1, \ldots, t_T\}$
- Input features $\mathbf{X} = \{\mathbf{X}_{z_1}, \ldots, \mathbf{X}_{z_k}\}$ and targets $\mathbf{Y} = \{\mathbf{Y}_{z_1}, \ldots, \mathbf{Y}_{z_k}\}$
- Target coverage level $1 - \epsilon$

**Find:** A forecasting function $f(\cdot)$ and prediction sets $\{\Gamma_{z_l}^{t_j}\}$ such that:

1. **(Accuracy)** $f(\cdot)$ produces accurate multi-source forecasts by modeling inter-source dependencies

2. **(Coverage)** For all sources $l \in \{1, \ldots, k\}$, time steps $j \in \{1, \ldots, T\}$, and samples $i \in \{1, \ldots, n\}$:

$$\mathbb{P}\big(y_{z_l}^{t_j(i)} \in \Gamma_{z_l}^{t_j}\big) \geq 1 - \epsilon \tag{1}$$

3. **(Efficiency)** Prediction sets $\Gamma_{z_l}^{t_j}$ are as tight as possible while satisfying coverage

**Key Challenges.** This problem presents two fundamental difficulties: (1) standard conformal prediction assumes exchangeability, which is violated by temporal dependencies; and (2) multi-source fusion introduces inter-source correlations that must be preserved during uncertainty aggregation. Our framework, CAPTAIN, can address both challenges through NIG-based uncertainty modeling and temporal copula calibration.

## 3  Theoretical Framework

This section establishes the theoretical foundation for integrating NIG distributions with temporal copulas under a conformal prediction framework. We first present NIG-based uncertainty modeling for multi-source fusion (Section 3.1), then introduce temporal copula calibration for valid coverage (Section 3.2), and finally prove end-to-end coverage guarantees (Section 3.3).

### 3.1  Multi-Source Uncertainty Modeling with NIG Distributions

We adopt Normal Inverse Gamma (NIG) distributions Ma et al. (2021) to model source-specific uncertainties. NIG distributions jointly capture prediction and uncertainty through parameters $(\delta, \gamma, \alpha, \beta)$ and crucially allow principled *evidence fusion* across sources. See Appendix A.2.3 for detailed NIG formulation.

**Definition 3.1** (NIG Evidence Fusion). Given two NIG distributions $\text{NIG}(\delta_1, \gamma_1, \alpha_1, \beta_1)$ and $\text{NIG}(\delta_2, \gamma_2, \alpha_2, \beta_2)$ representing independent evidence about shared latent parameters $(\mu, \sigma^2)$, their *evidence fusion* yields a new NIG distribution:

$$\text{NIG}(\delta_1, \gamma_1, \alpha_1, \beta_1) \oplus \text{NIG}(\delta_2, \gamma_2, \alpha_2, \beta_2) = \text{NIG}(\delta_*, \gamma_*, \alpha_*, \beta_*), \tag{2}$$

where the fused parameters are:

$$\gamma_* = \gamma_1 + \gamma_2 \tag{3}$$

$$\delta_* = \frac{\gamma_1 \delta_1 + \gamma_2 \delta_2}{\gamma_1 + \gamma_2} \tag{4}$$

$$\alpha_* = \alpha_1 + \alpha_2 - 1 \tag{5}$$

$$\beta_* = \beta_1 + \beta_2 + \frac{\gamma_1 \gamma_2}{\gamma_1 + \gamma_2}(\delta_1 - \delta_2)^2 / 2 \tag{6}$$

The fusion parameters have intuitive interpretations detailed in Appendix A.2.4. The following proposition establishes that NIG evidence fusion is well-behaved algebraically, enabling flexible multi-source combination.

**Proposition 3.2** (Properties of NIG Evidence Fusion). *The NIG evidence fusion operator $\oplus$ satisfies:*

1. **Commutativity:** $NIG_1 \oplus NIG_2 = NIG_2 \oplus NIG_1$

2. **Associativity:** $(NIG_1 \oplus NIG_2) \oplus NIG_3 = NIG_1 \oplus (NIG_2 \oplus NIG_3)$

3. **Identity:** $NIG(\delta, 0, 1, 0) \oplus NIG_1 = NIG_1$ *(uninformative prior)*

*Proof.* The proof follows by verifying commutativity and associativity through direct calculation of fusion parameters. See Appendix A.1.1 for complete details. □

**Multi-Source Fusion with Meta-Source** For $k$ sources, we apply NIG evidence fusion iteratively. Additionally, we introduce a learnable *meta-source* that captures inter-source dependencies not modeled by the independent fusion assumption.

**Definition 3.3** (Multi-Source NIG Fusion). Given source-specific NIG distributions $\{\text{NIG}_{z_l}\}_{l=1}^k$ and a meta-source $\text{NIG}_{\text{meta}}$, the fused distribution is:

$$\text{NIG}_{\text{fused}} = \text{NIG}_{\text{meta}} \oplus \bigoplus_{l=1}^k \text{NIG}_{z_l}. \tag{7}$$

By Proposition 3.2, the fusion order does not affect the result.

The meta-source serves a crucial role: while the $\oplus$ operator assumes sources provide independent evidence, real-world sources often exhibit correlations (e.g., ECG and blood pressure both respond to patient stress). The meta-source, learned from concatenated source representations, captures these shared patterns and adjusts the fused uncertainty accordingly.

### 3.2 Temporal Copula Calibration

While NIG distributions model uncertainty at each time step, they treat observations as independent. We address this through copula-based calibration Sun & Yu (2023), which models temporal dependencies in non-conformity scores. See Appendix A.2.5 for copula definitions and Sklar's theorem.

Let $(\delta_*^{t_j}, \alpha_*^{t_j}, \beta_*^{t_j})$ denote the fused NIG parameters at time step $t_j$, and define $\sigma_*^{t_j} = \sqrt{\beta_*^{t_j}/(\alpha_*^{t_j} - 1)}$ as the fused predictive standard deviation. The NIG-normalized non-conformity score is:

$$s^{t_j(i)} = \frac{|y^{t_j(i)} - \delta_*^{t_j(i)}|}{\sigma_*^{t_j(i)}}. \tag{8}$$

Define transformed scores $u^{(i)} = (u_1^{(i)}, \ldots, u_T^{(i)})$ where $u_j^{(i)} = \hat{F}_{t_j}(s^{t_j(i)})$ and $\hat{F}_{t_j}$ is the empirical CDF. The empirical copula is:

$$\hat{C}(\mathbf{u}) = \frac{1}{n} \sum_{i=1}^{n} \mathbf{1}\big[u^{(i)} \preceq \mathbf{u}\big], \tag{9}$$

using vector partial order $\mathbf{u} \preceq \mathbf{v} \iff \forall j : u_j \leq v_j$.

**Assumption 3.4** (Approximate Exchangeability). *The non-conformity score vectors $\{u^{(i)}\}_{i=1}^{n+1}$ (including the test sample) are approximately exchangeable within a sliding calibration window. Specifically, for any permutation $\pi$:*

$$\big(u^{(1)}, \ldots, u^{(n+1)}\big) \overset{d}{\approx} \big(u^{(\pi(1))}, \ldots, u^{(\pi(n+1))}\big). \tag{10}$$

Conditions under which this assumption holds are discussed in Appendix A.2.6.

### 3.3 Coverage Guarantee

We now state and prove our main result: that combining NIG evidence fusion with temporal copula calibration yields valid prediction intervals for multi-source time-series forecasting.

**Theorem 3.5** (Validity of CAPTAIN). *Under the following conditions:*

1. *The calibration and test samples satisfy Assumption 3.4 (approximate exchangeability);*

2. *The fused NIG parameters satisfy $\alpha_* > 1$, $\beta_* > 0$, $\gamma_* > 0$ (ensuring finite variance);*

3. *The empirical copula $\hat{C}$ is computed from $n$ calibration samples with $n \geq \lceil 1/\epsilon \rceil$;*

*the prediction sets $\Gamma^{t_j} = \{y : |y - \delta_*^{t_j}| \leq \sigma_*^{t_j} \cdot \hat{F}_{t_j}^{-1}(u_j^*)\}$ constructed by CAPTAIN from fused NIG parameters satisfy:*

$$\mathbb{P}\Big(\forall j \in \{1, \ldots, T\} : y^{t_j} \in \Gamma^{t_j}\Big) \geq 1 - \epsilon. \tag{11}$$

*Moreover, marginal coverage holds for each time step:*

$$\mathbb{P}\big(y^{t_j} \in \Gamma^{t_j}\big) \geq 1 - \epsilon. \tag{12}$$

We establish validity through a sequence of lemmas culminating in the main coverage guarantee.

**NIG Fusion Preserves Calibration Structure** The first key insight is that NIG evidence fusion operates deterministically on distribution parameters, preserving the exchangeability of samples.

**Lemma 3.6** (Fusion Preserves Sample Exchangeability). *Let $\{(\delta_k^{(i)}, \gamma_k^{(i)}, \alpha_k^{(i)}, \beta_k^{(i)})\}_{i=1}^{n+1}$ be exchangeable NIG parameter vectors for each source $k$. Then the non-conformity scores $\{s_{fused}^{(i)}\}_{i=1}^{n+1}$ computed from the fused NIG parameters are also exchangeable.*

*Proof.* The proof establishes that NIG fusion operates deterministically on parameters, preserving exchangeability through the composition of deterministic functions. See Appendix A.1.2 for complete details. □

**Copula-Based Quantile Function** Inspired by (Sun & Yu, 2023), we define a multivariate quantile function over the calibration set.

**Definition 3.7** (Empirical Multivariate Quantile). For target coverage $1 - \epsilon$ and calibration scores $\mathcal{U} = \{u^{(i)}\}_{i=1}^n$, define:

$$\hat{Q}_{1-\epsilon}(\mathcal{U}) = \underset{\mathbf{u}^* \in [0,1]^T}{\arg\min} \sum_{j=1}^{T} u_j^* \quad \text{s.t.} \quad \hat{C}(\mathbf{u}^*) \geq 1 - \epsilon, \tag{13}$$

where $\hat{C}$ is the empirical copula (Eq. 29).

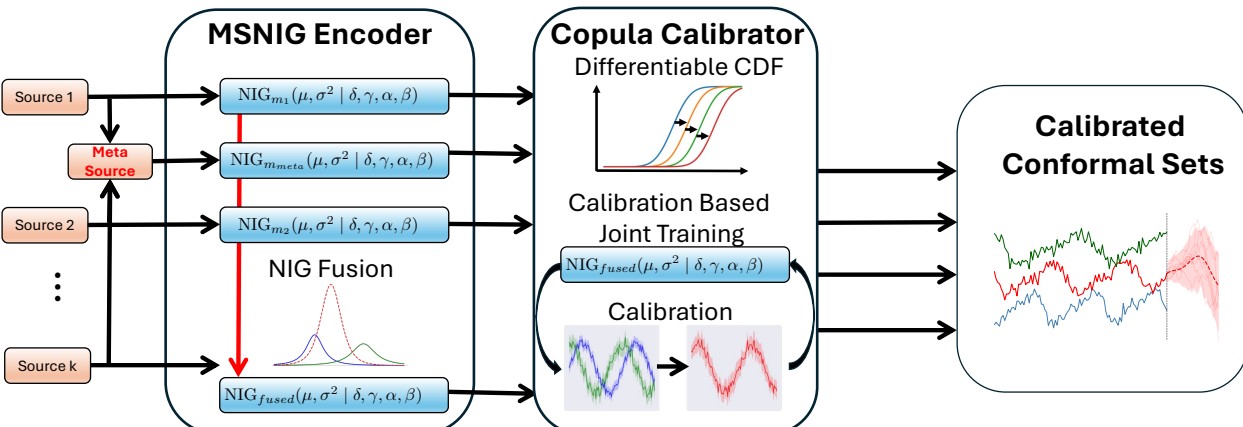

Figure 2: Overview of the CAPTAIN framework. **Stage 1 (MSNIG Encoder):** Source-specific encoders model uncertainties via NIG distributions, which are fused with a meta-source representation. **Stage 2 (Copula Calibrator):** Temporal copulas capture dependencies across the forecasting horizon, producing calibrated prediction intervals with coverage guarantees.

This quantile finds the smallest (in sum) threshold vector that covers at least $1 - \epsilon$ fraction of calibration samples jointly across all time steps.

**Coverage Proof**

*Proof of Theorem 3.5.* By Lemma 3.6, the NIG fusion operator preserves exchangeability of non-conformity scores, so the transformed score vectors $\{u^{(i)}\}_{i=1}^{n+1}$ satisfy approximate exchangeability under Assumption 3.4.

Under exchangeability, the rank of the test sample $u^{(n+1)}$ among the augmented calibration set $\mathcal{U} \cup \{u^{(n+1)}\}$ is uniformly distributed over $\{1, \ldots, n+1\}$. This yields the conformal prediction bound:

$$\mathbb{P}\left[u^{(n+1)} \preceq \hat{Q}_{1-\epsilon}(\mathcal{U})\right] \geq \frac{\lceil (1-\epsilon)(n+1) \rceil}{n+1} \geq 1 - \epsilon. \tag{14}$$

The vector partial order $u^{(n+1)} \preceq \mathbf{u}^*$ holds if and only if $u_j^{(n+1)} \leq u_j^*$ for all $j \in \{1, \ldots, T\}$, which is equivalent to $y^{t_j} \in \Gamma^{t_j}$ for all $j$, establishing joint coverage. Marginal coverage follows immediately since joint coverage implies marginal coverage. See Appendix A.1.3 for detailed calculations and intermediate steps. $\square$

Additional remarks on computational complexity and individual source predictions are provided in Appendix A.2.8.

## 4 The CAPTAIN Framework

We present CAPTAIN, a two-stage framework for uncertainty quantification in multi-source time-series forecasting. The first stage employs Normal Inverse Gamma (NIG) distributions to model source-specific and inter-source uncertainties, adapting to variations in data quality across heterogeneous sources. The second stage leverages temporal copulas to capture dependencies across the forecasting horizon, ensuring structured uncertainty propagation with provable coverage guarantees. Together, these components provide theoretically grounded and robust uncertainty estimates suitable for real-world applications.

### 4.1 Framework Overview

Existing uncertainty quantification methods for time-series forecasting face two key limitations: (1) Gaussian-based approaches capture only first and second moments, providing limited expressiveness for heteroscedastic uncertainty; (2) sampling-based Bayesian methods incur significant computational overhead and scale poorly with multiple data sources. In contrast, NIG distributions offer closed-form expressions for both aleatoric and epistemic uncertainties while enabling principled fusion across sources, both critical properties for multi-source settings. As illustrated in Figure 2, CAPTAIN consists of two complementary modules:

**1. Multi-Source NIG Encoder (MSNIG):** This module addresses *Q1 (Inter-Source Temporal Uncertainty)* by modeling source-specific uncertainties using NIG distributions and incorporating a meta-source representation to capture inter-source dependencies. The NIG summation operator enables principled uncertainty fusion that adapts to varying data quality across sources.

**2. Copula-Based Temporal Calibrator:** This module addresses *Q2 (Theoretical Guarantees)* by applying temporal copulas to model uncertainty evolution across the forecasting horizon. A differentiable calibration mechanism ensures that prediction intervals achieve the target coverage level while respecting temporal dependencies.

## 4.2 Multi-Source NIG Encoder (MSNIG)

The MSNIG encoder captures uncertainty at three levels: (i) source-specific uncertainty estimation, (ii) meta-source learning for inter-source dependencies, and (iii) principled uncertainty fusion via NIG summation.

**Source-Specific NIG Modeling** For each source $l \in \{1, \ldots, K\}$ and time step $j \in \{1, \ldots, T\}$, we model the predictive distribution using a hierarchical NIG prior that captures both aleatoric (data) and epistemic (model) uncertainties.

**Observation Model.** Each prediction follows a Gaussian distribution with unknown mean and variance:

$$y_{l,j}^{(i)} \sim \mathcal{N}\big(\mu_{l,j}^{(i)}, (\sigma_{l,j}^{(i)})^2\big), \tag{15}$$

where $i \in \{1, \ldots, n\}$ indexes samples.

**Hierarchical Prior.** To quantify uncertainty over the unknown parameters, we place conjugate priors:

$$\mu_{l,j}^{(i)} \sim \mathcal{N}\big(\delta_{l,j}, (\sigma_{l,j}^{(i)})^2/\gamma_{l,j}\big), \tag{16}$$

$$(\sigma_{l,j}^{(i)})^2 \sim \text{Inv-Gamma}(\alpha_{l,j}, \beta_{l,j}), \tag{17}$$

where $\boldsymbol{\theta}_{l,j} = (\delta_{l,j}, \gamma_{l,j}, \alpha_{l,j}, \beta_{l,j})$ are the NIG parameters output by the source-specific encoder $f_l$:

$$\boldsymbol{\theta}_{l,j} = f_l(\mathbf{x}_{l,1:j}), \tag{18}$$

with $\mathbf{x}_{l,1:j}$ denoting the input sequence for source $l$ up to time $j$.

**Interpretation of NIG Parameters:**

- $\delta$: Location parameter (predicted mean)
- $\gamma$: Precision of the mean estimate (higher $\gamma$ indicates greater confidence)
- $\alpha, \beta$: Shape and scale of the variance distribution (epistemic uncertainty)

**Meta-Source Learning** To capture inter-source dependencies that individual encoders may miss, we introduce a meta-source that processes a joint representation of all sources:

$$\mathbf{x}_{\text{meta}} = g_{\text{fuse}}\big([\mathbf{x}_1, \ldots, \mathbf{x}_K]\big), \tag{19}$$

where $g_{\text{fuse}}(\cdot)$ is a learnable fusion function (e.g., concatenation followed by a linear projection). The meta-source encoder outputs its own NIG parameters:

$$\boldsymbol{\theta}_{\text{meta},j} = (\delta_{\text{meta},j}, \gamma_{\text{meta},j}, \alpha_{\text{meta},j}, \beta_{\text{meta},j}) = f_{\text{meta}}(\mathbf{x}_{\text{meta},1:j}). \tag{20}$$

**Meta-Source Learning and NIG Fusion.** The meta-source aggregates representations across all sources to serve as a global uncertainty estimator that complements source-specific predictions. The NIG summation operator (Definition 3.1) automatically weights each source by its precision, downweighting unreliable sources. Detailed discussion on meta-source benefits and fusion properties is provided in Appendix A.4.1.

Applying the summation operator across all $K$ sources and the meta-source:

$$\text{NIG}_{\text{fused},j} = \text{NIG}_{\text{meta},j} \bigoplus_{l=1}^{K} \text{NIG}_{l,j}. \tag{21}$$

The fused distribution maintains the NIG form with parameters:

$$\delta_{\text{fused},j} = \frac{\sum_{l=0}^{K} \gamma_{l,j} \delta_{l,j}}{\sum_{l=0}^{K} \gamma_{l,j}}, \quad \gamma_{\text{fused},j} = \sum_{l=0}^{K} \gamma_{l,j},$$

$$\alpha_{\text{fused},j} = \sum_{l=0}^{K} \alpha_{l,j} - K, \quad \beta_{\text{fused},j} = \sum_{l=0}^{K} \beta_{l,j} + \frac{1}{2} \sum_{l=0}^{K} \gamma_{l,j} (\delta_{l,j} - \delta_{\text{fused},j})^2, \tag{22}$$

where index $l = 0$ corresponds to the meta-source.

**NIG Loss Function** The MSNIG encoder is trained by minimizing the negative log-likelihood under the NIG model. For the fused distribution at time step $j$:

$$\mathcal{L}_{\text{NIG},j} = \frac{1}{2} \log \left( \frac{\pi}{\gamma_j} \right) - \alpha_j \log(\Omega_j) + \left( \alpha_j + \frac{1}{2} \right) \log \left( (y_j - \delta_j)^2 \gamma_j + \Omega_j \right) + \log \Psi_j, \tag{23}$$

where $\Omega_j = 2\beta_j(1 + \gamma_j)$ and $\Psi_j = \Gamma(\alpha_j)/\Gamma(\alpha_j + \frac{1}{2})$. We omit the "fused" subscript for brevity.

The total NIG loss aggregates over all time steps and components:

$$\mathcal{L}_{\text{NIG}} = \sum_{j=1}^{T} \left( \sum_{l=1}^{K} \mathcal{L}_{\text{NIG},l,j} + \mathcal{L}_{\text{NIG,meta},j} + \mathcal{L}_{\text{NIG,fused},j} \right). \tag{24}$$

### 4.3 Copula-Based Temporal Calibrator

While the MSNIG encoder provides well-structured uncertainty estimates, these estimates may not satisfy formal coverage guarantees due to model misspecification or distribution shift. The copula calibrator addresses this by constructing prediction intervals with provable coverage on temporal horizon.

**Novelty over Prior Work.** While conformal prediction Vovk et al. (2005) and copulas Messoudi et al. (2021); Sun & Yu (2023) have been studied independently, CAPTAIN is the first to integrate: (i) multi-source NIG-based uncertainty fusion, (ii) temporal copula calibration, and (iii) differentiable end-to-end training with formal coverage guarantees.

**Prediction Set Construction** For each component $c \in \{1, \ldots, K, \text{meta}, \text{fused}\}$, we construct prediction sets $\Gamma_c = \{\Gamma_{c,1}, \ldots, \Gamma_{c,T}\}$ satisfying:

$$\mathbb{P}(y_{c,j}^{(i)} \in \Gamma_{c,j}) \geq 1 - \epsilon, \quad \forall j \in \{1, \ldots, T\}. \tag{25}$$

**Non-Conformity Scores and Empirical CDF** For each component $c$ and time step $j$, we compute *NIG-normalized* non-conformity scores:

$$s_{c,j}^{(i)} = \frac{|y_j^{(i)} - \delta_{c,j}^{(i)}|}{\sigma_{c,j}^{(i)}}, \quad \text{where} \quad \sigma_{c,j} = \sqrt{\frac{\beta_{c,j}}{\alpha_{c,j} - 1} \cdot \left( 1 + \frac{1}{\gamma_{c,j}} \right)}, \tag{26}$$

where $\delta_{c,j}^{(i)}$ is the predicted mean and $\sigma_{c,j}^{(i)}$ is the NIG-derived scale.

**Design Rationale.** Unlike prior conformal methods using raw residuals, CAPTAIN normalizes scores by fused NIG uncertainty, making scores comparable across time steps and incorporating multi-source fusion quality into calibration. See Appendix A.4.2 for detailed justification.

The empirical CDF over a calibration buffer of size $n$ is:

$$\hat{F}_{c,j}(s) = \frac{1}{n+1} \left( 1 + \sum_{i=1}^{n} \mathbb{1}\{s_{c,j}^{(i)} \leq s\} \right). \tag{27}$$

We maintain a rolling calibration matrix of transformed scores:

$$\mathbf{U}_c = \{\mathbf{u}_c^{(1)}, \ldots, \mathbf{u}_c^{(n)}\}, \quad \mathbf{u}_c^{(i)} = \big(\hat{F}_{c,1}(s_{c,1}^{(i)}), \ldots, \hat{F}_{c,T}(s_{c,T}^{(i)})\big). \tag{28}$$

**Empirical Copula** To capture temporal dependencies in the non-conformity scores, we compute the empirical copula :

$$\hat{C}(\mathbf{u}_c) = \frac{1}{n} \sum_{i=1}^{n} \prod_{j=1}^{T} \mathbb{1}\{u_{c,j}^{(i)} < u_{c,j}\}. \tag{29}$$

The goal is to find threshold vectors $\mathbf{u}_c^* = (u_{c,1}^*, \ldots, u_{c,T}^*)$ such that $\hat{C}(\mathbf{u}_c^*) \geq 1 - \epsilon$, formulated as:

$$\min_{\mathbf{u}_c^*} \sum_{j=1}^{T} u_{c,j}^* \quad \text{s.t.} \quad \hat{C}(\mathbf{u}_c^*) \geq 1 - \epsilon. \tag{30}$$

**Differentiable Calibration** To enable end-to-end training, we introduce learnable calibration parameters $\{\tau_j\}_{j=1}^{T}$ and replace the hard indicator functions with smooth sigmoid approximations:

$$\hat{P}_{\text{coverage}} = \frac{1}{n} \sum_{i=1}^{n} \left[ \prod_{j=1}^{T} \sigma\big((\tau_j - \hat{F}_{c,j}(s_{c,j}^{(i)})) \cdot \kappa\big) \right], \tag{31}$$

where $\sigma(\cdot)$ is the sigmoid function and $\kappa > 0$ is a temperature parameter controlling the approximation sharpness (we use $\kappa = 1000$ in practice).

**Coverage Loss** The coverage loss encourages the empirical coverage to match the target level $1 - \epsilon$:

$$\mathcal{L}_{\text{COV}} = \left| \frac{1}{n \cdot C \cdot T} \sum_{i=1}^{n} \sum_{c=1}^{C} \prod_{j=1}^{T} \mathbb{1}\left\{ u_{c,j}^{(i)} < \hat{F}_{c,j}(s_{c,j}) \right\} - (1 - \epsilon) \right|, \tag{32}$$

where $C = K + 2$ is the total number of components (K sources + meta + fused).

This calibration mechanism provides four key properties: (i) coverage guarantees via learned thresholds $\tau_j$ that enforce target coverage across time steps; (ii) temporal coherence through the copula structure, which captures dependencies and prevents overly conservative intervals; (iii) adaptive calibration via the rolling buffer that accommodates distribution shifts; and (iv) end-to-end differentiability through sigmoid relaxation, enabling joint optimization with the NIG encoder.

### 4.4 Joint Optimization

The complete CAPTAIN framework is trained end-to-end by minimizing a unified objective that balances uncertainty estimation accuracy and coverage calibration:

$$\mathcal{L} = \underbrace{\sum_{l=1}^{K} \sum_{j=1}^{T} \mathcal{L}_{\text{NIG},l,j} + \sum_{j=1}^{T} \mathcal{L}_{\text{NIG,meta},j} + \sum_{j=1}^{T} \mathcal{L}_{\text{NIG,fused},j}}_{\text{Uncertainty Estimation}} + \lambda \underbrace{\sum_{c=1}^{C} \mathcal{L}_{\text{COV},c}}_{\text{Coverage Calibration}}, \tag{33}$$

where $\lambda > 0$ balances uncertainty estimation and coverage calibration. During training, each forward pass computes NIG parameters for all sources, the meta-source, and the fused representation. The rolling calibration buffer is updated with current non-conformity scores, and all parameters are optimized jointly via gradient descent. This joint optimization enables CAPTAIN to simultaneously learn accurate uncertainty representations and maintains valid coverage guarantees, providing coherent predictions across sources and time steps.

Table 1: Coverage (%) and interval width at 90% target. ✓= valid (≥90%), × = invalid. **CAPTAIN is the only method achieving valid coverage on all 5 datasets.**

| Method | Synthetic | | Shaoxing (ECG) | | Air Quality | | NGSIM (Traffic) | | ETTh1 | | Valid |
| | Cov. | Width | Cov. | Width | Cov. | Width | Cov. | Width | Cov. | Width | Count |
|---|---|---|---|---|---|---|---|---|---|---|---|
| MC-Dropout | $89.9_{\pm1.4}$× | 2.54 | $91.7_{\pm4.3}$✓ | 2.77 | $88.9_{\pm4.7}$× | 1.98 | $91.7_{\pm1.8}$✓ | 3.05 | $85.3_{\pm6.2}$× | 5.47 | 2/5 |
| BJ-LSTM | $89.9_{\pm1.0}$× | 4.29 | $87.6_{\pm5.1}$× | 4.26 | $87.7_{\pm2.5}$× | 4.20 | $90.4_{\pm2.1}$✓ | 5.15 | $88.1_{\pm3.4}$× | 6.77 | 1/5 |
| CF-LSTM | $90.4_{\pm4.7}$✓ | 2.15 | $90.9_{\pm1.5}$✓ | 2.36 | $87.1_{\pm4.6}$× | 1.76 | $88.5_{\pm5.3}$× | 2.53 | $88.4_{\pm1.9}$× | 5.14 | 2/5 |
| CopulaCPTS | $89.2_{\pm4.4}$× | 2.13 | $89.9_{\pm0.7}$× | 2.32 | $84.8_{\pm4.6}$× | 1.74 | $87.7_{\pm5.5}$× | 2.51 | $88.2_{\pm2.4}$× | 5.09 | 0/5 |
| SPCI | $90.6_{\pm2.5}$✓ | 2.22 | $89.5_{\pm2.5}$× | 2.23 | $85.9_{\pm2.5}$× | 1.79 | $87.7_{\pm2.5}$× | 2.51 | $85.8_{\pm2.5}$× | 5.37 | 1/5 |
| MultiDimSPCI | $92.1_{\pm2.0}$✓ | 2.27 | $93.7_{\pm2.0}$✓ | 2.65 | $88.2_{\pm2.0}$× | 1.88 | $90.3_{\pm2.0}$✓ | 2.71 | $87.6_{\pm2.0}$× | 5.51 | 3/5 |
| CPTC | $89.3_{\pm2.2}$× | 2.41 | $90.9_{\pm2.2}$✓ | 2.68 | $91.5_{\pm2.2}$✓ | 2.05 | $93.5_{\pm2.2}$✓ | 2.74 | $93.4_{\pm2.2}$✓ | 5.63 | 4/5 |
| CAPTAIN (Ours) | $\mathbf{93.3_{\pm3.8}}$✓ | **2.14** | $\mathbf{94.1_{\pm3.1}}$✓ | **2.31** | $\mathbf{92.6_{\pm3.6}}$✓ | **1.90** | $\mathbf{94.7_{\pm4.6}}$✓ | **2.64** | $\mathbf{92.8_{\pm0.4}}$✓ | **5.39** | **5/5** |

## 5 Experiments

We evaluate CAPTAIN on synthetic and real-world multi-source time-series datasets, addressing three research questions:

**Q1 Calibration:** Does CAPTAIN achieve valid coverage while maintaining efficient prediction intervals?

**Q2 Component Analysis:** How do the meta-source, NIG fusion, and copula calibration contribute to performance?

**Q3 Robustness:** Does CAPTAIN maintain valid coverage under both in-distribution and out-of-distribution conditions?

### 5.1 Experimental Setup

**Datasets.** We evaluate on five datasets with standard splits (60% train, 10% validation, 10% calibration, 20% test). *Synthetic*: three sources with inter-source correlation $\rho = 0.5$ and configurable noise (input: 48, horizon: 12). *Shaoxing (ECG)*: 12-lead physiological signals with high temporal autocorrelation ($\rho = 0.8$). *Air Quality*: multi-source environmental sensors for pollution prediction. *NGSIM (Traffic)*: vehicle trajectory data with position/velocity measurements. *ETTh1* Zhou et al. (2021): electricity transformer temperature (17,301 samples, input: 96, horizon: 24), testing scalability on industrial data.

**Baselines.** We compare against seven methods spanning three paradigms: *Conformal prediction*—CF-LSTM Stankeviciute et al. (2021), CopulaCPTS Sun & Yu (2023), SPCI Xu & Xie (2023), MultiDim-SPCI Xu et al. (2024b), and CPTC Sun & Yu (2025); *Frequentist*—BJ-LSTM Alaa & Van Der Schaar (2020); *Bayesian*—MC-Dropout Gal & Ghahramani (2016).

**Metrics.** Coverage measures the fraction of true values within predicted intervals (valid if $\geq$ 90%). Interval width quantifies efficiency (smaller is better), conditional on valid coverage.

**Implementation.** All models use LSTM encoders with a hidden dimension of 64. For fair comparison, each baseline uses multi-source LSTMs (one per source) with the same hidden dimension, differing only in their uncertainty quantification mechanism: MC-Dropout uses dropout sampling, BJ-LSTM uses block jackknife, and conformal methods (CF-LSTM, CopulaCPTS, SPCI, MultiDimSPCI, CPTC) wrap the base predictor with calibration-based intervals. CAPTAIN uses the same LSTM backbone but outputs NIG parameters instead of point predictions, with additional meta-source fusion and copula calibration—these are the key contributions being evaluated. Results are reported as mean ± standard deviation across 5 independent runs. Target coverage: 90%.

### 5.2 Main Results (Q1: Calibration)

Table 1 presents coverage and interval width across all datasets. We highlight three findings addressing **Q1**.

**Finding 1: CAPTAIN is the only method achieving valid coverage on all datasets.** CPTC achieves 4/5, MultiDimSPCI 3/5, while CopulaCPTS fails on all datasets (0/5) despite using temporal copulas. This demonstrates that copulas alone are insufficient—structured multi-source uncertainty fusion is essential for consistent calibration.

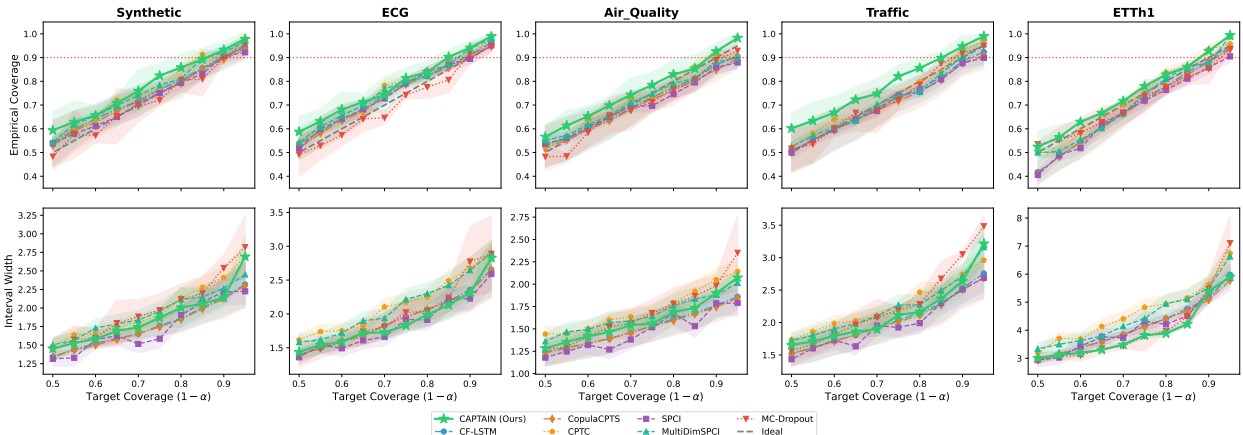

Figure 3: Calibration (top) and interval width (bottom) across target coverage levels. Dashed diagonal: ideal; red line: 90% threshold. CAPTAIN tracks ideal calibration with minimal variance.

**Finding 2: CAPTAIN achieves the tightest intervals among valid methods.** Conditional on valid coverage, CAPTAIN produces the narrowest intervals on all 5 datasets, with an average width reduction of 3.4% compared to the best valid baseline per dataset. On datasets where both CAPTAIN and CPTC achieve valid coverage, CAPTAIN produces 7.3% narrower intervals on average.

**Finding 3: CAPTAIN maintains calibration across coverage levels.** Figure 3 shows calibration curves for $1 - \alpha \in [0.50, 0.95]$. CAPTAIN consistently tracks the ideal diagonal, while baselines either undercover (CopulaCPTS, SPCI) or exhibit high variance (MC-Dropout).

### 5.3 Ablation Studies (Q2: Component Analysis)

Table 2 presents systematic ablation results on the Synthetic dataset, isolating the contribution of each component.

**Copula calibration is essential.** Removing copula calibration (w/o Calibration) catastrophically drops coverage to 72.0%, failing to achieve valid uncertainty quantification despite producing 19.5% narrower intervals (1.578 vs. 1.959). This demonstrates that raw NIG intervals severely undercover—the conformal calibration step is indispensable for valid coverage guarantees.

**Meta-source improves efficiency.** Without meta-source fusion, the model maintains valid coverage (90.1%) but intervals widen significantly by 13.5% (2.224 vs. 1.959). The meta-source captures cross-source dependencies that individual source models miss, enabling tighter intervals while preserving calibration.

**NIG fusion outperforms naive averaging.** Replacing NIG evidence fusion (Eq. 3–6) with simple mean pooling increases interval width by 5.8% (2.072 vs. 1.959) while slightly overcovering (92.2%). The precision-weighted fusion mechanism (Eq. 4) effectively down-weights uncertain sources, producing tighter and better-calibrated intervals.

See Appendix A.6.1 for extended ablation analysis including non-conformity score design comparisons and coverage-width trade-off visualizations across all datasets.

Table 2: Ablation study (Synthetic, target: 90%). $\times$ = invalid coverage.

| Configuration | Coverage (%) | Width | $\Delta$ Width |
|---|---|---|---|
| CAPTAIN (Full) | **91.0**✓ | **1.959** | – |
| w/o Meta-source | 90.1✓ | 2.224 | +13.5% |
| w/o NIG-Fusion | 92.2✓ | 2.072 | +5.8% |
| w/o Calibration | 72.0× | 1.578 | -19.5% |

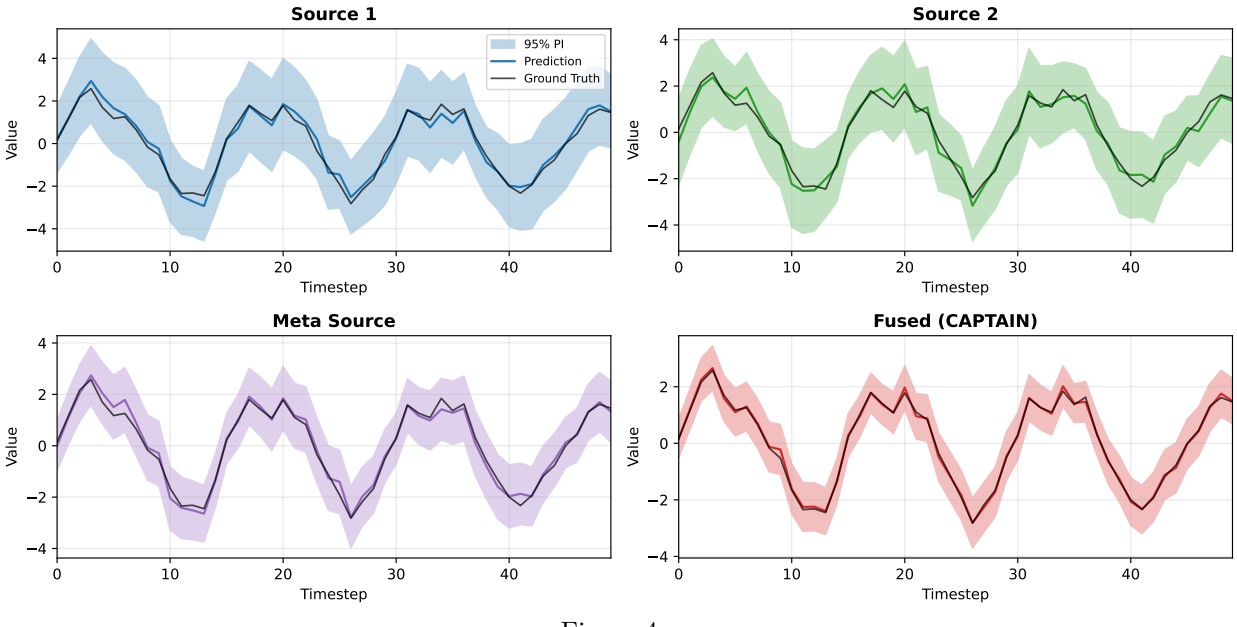

Figure 4:

## 5.4 Robustness Analysis (Q3: Robustness)

We analyze CAPTAIN's robustness under distribution shift by categorizing datasets based on calibration-test similarity.

**In-distribution performance.** On Synthetic and Shaoxing, where test data closely matches calibration, CAPTAIN achieves 93.3% and 94.1% coverage with low variance ($\pm 3.8$, $\pm 3.1$). Baselines also achieve valid coverage here, confirming standard conformal methods work when exchangeability holds.

**Out-of-distribution robustness.** On Air Quality and ETTh1, temporal distribution shift causes most baselines to fail: CF-LSTM (87.1%, 88.4%), MultiDimSPCI (88.2%, 87.6%), and MC-Dropout (88.9%, 85.3%) all undercover. CAPTAIN maintains valid coverage (92.6%, 92.8%) via precision-weighted fusion (Eq. 4), which down-weights unreliable sources under shift. NGSIM Traffic exhibits moderate shift; CAPTAIN (94.7%) outperforms CF-LSTM (88.5%) and SPCI (87.7%).

**Key insight.** CopulaCPTS's complete failure (0/5) despite using temporal copulas confirms that temporal modeling alone is insufficient as multi-source uncertainty fusion is essential. CAPTAIN achieves 5/5 valid coverage vs. 0–4/5 for baselines with modest overhead: $\sim 10\%$ inference, $\sim 18\%$ training (ETTh1: 45.2s vs. 38.1s/epoch).

# 6 Conclusion

We addressed two key challenges in multi-source time-series uncertainty quantification: (1) capturing joint uncertainty across sources and (2) ensuring valid theoretical coverage over time. We proposed CAPTAIN, a two-stage framework that integrates NIG distributions for principled multi-source uncertainty fusion with temporal copulas for conformal calibration. Experiments on five diverse datasets (Synthetic, Shaoxing ECG, Air Quality, NGSIM Traffic, and ETTh1) demonstrate that CAPTAIN is the **only method achieving valid coverage ($\geq$90%) on all datasets** (92.6–94.7%), compared against seven baselines including CF-LSTM, CopulaCPTS, CPTC, SPCI, MultiDimSPCI (ICML 2024), BJ-LSTM, and MC-Dropout. On the challenging ETTh1 dataset, CAPTAIN achieves 92.8% coverage where CF-LSTM (88.4%) and MultiDimSPCI (87.6%) fail, with **4.3% tighter intervals** than the next valid baseline CPTC (5.39 vs 5.63). Ablation studies confirm the complementary contributions of meta-source learning, NIG evidence fusion, and copula-based calibration. Future work includes extending CAPTAIN to irregularly sampled time series and incorporating attention-based architectures for longer forecast horizons.

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

# A  Appendix

This appendix provides complete proofs, theoretical analysis, experimental details, and related work. Organization:

- Section A.1: Proofs of theoretical results (Proposition 3.2, Lemma 3.6, Theorem 3.5)

- Section A.2: Theoretical foundations (time-series forecasting, conformal prediction, NIG distributions, fusion parameter interpretations, copulas, exchangeability conditions, computational complexity)

- Section A.3: Discussion of assumptions, connections to $\beta$-NLL, and independence mitigation

- Section A.4: Algorithm implementation and design choices (meta-source architecture, non-conformity score design)

- Section A.5: Dataset target variables and source configurations

- Section A.6: Experimental details and analysis (ablation study, score design comparison)

- Section A.7: Robustness and sensitivity analysis (joint vs. sequential training, inter-source correlation, heavy-tailed noise, exchangeability diagnostics, fully-integrated copula, $\beta$-NLL, multi-target channel cycling, look-ahead forecasting)

- Section A.9: Baseline failure analysis, computational benchmarks, and CAPTAIN's failure modes

- Section A.10: Related work on multi-source forecasting, uncertainty quantification, and conformal prediction

## A.1  Proofs of Theoretical Results

### A.1.1  Proof of Proposition 3.2

*Proof.* **Commutativity.** Direct from symmetry in Eqs. (3)–(6).

**Associativity.** For three sources, let $\mathrm{NIG}_{12} = \mathrm{NIG}_1 \oplus \mathrm{NIG}_2$. Then:

$$\gamma_{123} = \gamma_{12} + \gamma_3 = (\gamma_1 + \gamma_2) + \gamma_3 = \gamma_1 + \gamma_2 + \gamma_3,$$
$$\delta_{123} = \frac{\gamma_{12}\delta_{12} + \gamma_3\delta_3}{\gamma_{123}} = \frac{\gamma_1\delta_1 + \gamma_2\delta_2 + \gamma_3\delta_3}{\gamma_1 + \gamma_2 + \gamma_3}.$$

Symmetric in all indices. Similar calculations confirm associativity for $\alpha_*$ and $\beta_*$.

**Identity.** $\mathrm{NIG}(\delta, 0, 1, 0) \oplus \mathrm{NIG}_1 = \mathrm{NIG}_1$ since $\gamma = 0$ contributes no precision and $\alpha = 1, \beta = 0$ represents an uninformative prior. $\square$

### A.1.2  Proof of Lemma 3.6

*Proof.* The NIG fusion operator (Definition 3.1) computes fused parameters deterministically:

$$(\delta_*^{(i)}, \gamma_*^{(i)}, \alpha_*^{(i)}, \beta_*^{(i)}) = f\big((\delta_1^{(i)}, \gamma_1^{(i)}, \alpha_1^{(i)}, \beta_1^{(i)}), \ldots, (\delta_K^{(i)}, \gamma_K^{(i)}, \alpha_K^{(i)}, \beta_K^{(i)})\big). \tag{34}$$

Non-conformity scores are:

$$s_{\mathrm{fused}}^{(i)} = \frac{|y^{(i)} - \delta_*^{(i)}|}{\sqrt{\beta_*^{(i)}/(\alpha_*^{(i)} - 1)}}. \tag{35}$$

Since $f$ and score computation are deterministic functions applied identically to all samples, any permutation $\pi$ induces the same permutation of fused scores. By exchangeability of source NIG parameters and determinism, fused scores inherit exchangeability. $\square$

### A.1.3 Proof of Theorem 3.5

*Proof.* We prove Eq. (11); Eq. (12) follows as corollary.

Let $u^{(n+1)}$ be the transformed score vector for the test sample. By Lemma 3.6 and Assumption 3.4, the augmented set $\mathcal{U} \cup \{u^{(n+1)}\}$ is exchangeable.

*Rank distribution.* Under exchangeability, the rank of $u^{(n+1)}$ among $\mathcal{U} \cup \{u^{(n+1)}\}$ is uniformly distributed over $\{1, \ldots, n+1\}$. Following (Vovk et al., 2005):

$$\mathbb{P}\big[u^{(n+1)} \preceq \hat{Q}_{1-\epsilon}(\mathcal{U} \cup \{\infty\})\big] = \frac{\lceil(1-\epsilon)(n+1)\rceil}{n+1} \geq 1 - \epsilon, \tag{36}$$

where $\infty = (\infty, \ldots, \infty)$.

*Quantile equivalence.* Adding $\infty$ does not change the quantile for finite thresholds:

$$\hat{Q}_{1-\epsilon}(\mathcal{U} \cup \{\infty\}) = \hat{Q}_{1-\epsilon}(\mathcal{U} \cup \{u^{(n+1)}\}), \tag{37}$$

since $\infty \succ \mathbf{u}^*$ for any finite $\mathbf{u}^*$.

*Joint coverage.* By vector partial order definition $\mathbf{u} \preceq \mathbf{v} \iff \forall j : u_j \leq v_j$:

$$u^{(n+1)} \preceq \mathbf{u}^* \iff \forall j \in \{1, \ldots, T\} : u_j^{(n+1)} \leq u_j^*. \tag{38}$$

Converting to prediction sets: $u_j^{(n+1)} \leq u_j^*$ iff $s^{t_j(n+1)} \leq \hat{F}_{t_j}^{-1}(u_j^*)$ iff $y^{t_j(n+1)} \in \Gamma^{t_j}$.

Therefore:

$$\mathbb{P}\Big(\forall j : y^{t_j(n+1)} \in \Gamma^{t_j}\Big) = \mathbb{P}\big[u^{(n+1)} \preceq \mathbf{u}^*\big] \geq 1 - \epsilon. \tag{39}$$

Marginal coverage follows since joint coverage implies marginal coverage:

$$\mathbb{P}\big(y^{t_j} \in \Gamma^{t_j}\big) \geq \mathbb{P}\Big(\forall j' : y^{t_{j'}} \in \Gamma^{t_{j'}}\Big) \geq 1 - \epsilon. \tag{40}$$

$\square$

## A.2 Theoretical Foundations

### A.2.1 Time-Series Forecasting Background

Time-series forecasting predicts future values based on historical observations. Classical methods such as ARIMA (Box & Jenkins, 1968) employ statistical modeling, while modern deep learning approaches such as RNNs (Salinas et al., 2020), LSTMs (Hochreiter, 1997; Graves & Graves, 2012), and Transformers (Zhou et al., 2021; Zeng et al., 2023) capture complex temporal dependencies through learned representations.

**Single-Source Forecasting.** Given a look-back window $w$, single-source models predict:

$$\hat{y}^{t+1} = f\big(y^{t-w:t}, \mathbf{x}^{t-w:t}\big), \tag{41}$$

where $y^{t-w:t}$ and $\mathbf{x}^{t-w:t}$ denote historical targets and input features, and $f(\cdot)$ is the learned forecasting function.

**Multi-Source Forecasting.** Real-world applications often involve multiple related data sources capturing different aspects of the same phenomenon (AlSaad et al., 2024; Zonta et al., 2022). Multi-source models (Afyouni et al., 2022; Yang et al., 2017) extend this by learning source-specific representations and fusing them:

$$\hat{y}^{t+1} = g\Big(f_{z_1}(y_{z_1}^{t-w:t}, \mathbf{x}_{z_1}^{t-w:t}), \ldots, f_{z_k}(y_{z_k}^{t-w:t}, \mathbf{x}_{z_k}^{t-w:t})\Big), \tag{42}$$

where $f_{z_l}(\cdot)$ extracts source-specific features and $g(\cdot)$ fuses information across sources.

### A.2.2 Conformal Prediction Background

Uncertainty quantification (UQ) methods provide confidence measures alongside point predictions. Two dominant paradigms exist: *Bayesian methods* (Wu et al., 2021; Lakshminarayanan et al., 2017) model uncertainty through posterior distributions over parameters, while *Frequentist methods* (Alaa & Van Der Schaar, 2020; Park et al., 2022) construct intervals based on sampling variability. Both require distributional assumptions that may not hold in practice.

Conformal prediction (CP) (Vovk et al., 2005) provides a distribution-free alternative with finite-sample coverage guarantees. Given a calibration set and a non-conformity score function, CP constructs prediction sets guaranteed to contain the true outcome with probability at least $1 - \epsilon$.

**Theorem A.1** (Split Conformal Prediction (Vovk et al., 2005)). *Let $\{(\mathbf{x}_i, y_i)\}_{i=1}^n$ be exchangeable calibration samples and $(\mathbf{x}_{n+1}, y_{n+1})$ a test point. For a score function $s : \mathcal{X} \times \mathcal{Y} \to \mathbb{R}$ and target coverage $1 - \epsilon \in (0, 1)$, define:*

$$\Gamma(\mathbf{x}_{n+1}) = \{y : s(\mathbf{x}_{n+1}, y) \leq q_{1-\epsilon}\}, \tag{43}$$

*where $q_{1-\epsilon}$ is the $\lceil(n+1)(1-\epsilon)\rceil$-th smallest value in $\{s(\mathbf{x}_i, y_i)\}_{i=1}^n$. Then:*

$$\mathbb{P}(y_{n+1} \in \Gamma(\mathbf{x}_{n+1})) \geq 1 - \epsilon. \tag{44}$$

The scores $\{s(\mathbf{x}_i, y_i)\}_{i=1}^n$, termed *non-conformity scores*, measure how atypical each observation is relative to the model's predictions. Common choices include absolute residuals $|y - \hat{y}|$ for regression and softmax-based scores for classification (Romano et al., 2020; Angelopoulos et al., 2020).

### A.2.3 Normal Inverse Gamma Distributions

We adopt Normal Inverse Gamma (NIG) distributions (Ma et al., 2021) to model source-specific uncertainties. Unlike Gaussian assumptions modeling only predictive mean, NIG distributions jointly capture prediction and uncertainty through a hierarchical prior.

**Definition.** For source $z_l$ at time $t_j$:

$$y_{z_l}^{t_j} \sim \mathcal{N}(\mu, \sigma^2), \quad \mu \sim \mathcal{N}(\delta, \sigma^2/\gamma), \quad \sigma^2 \sim \text{Inv-Gamma}(\alpha, \beta), \tag{45}$$

where $(\delta, \gamma, \alpha, \beta)$ are NIG parameters. Expected values:

$$\mathbb{E}[\mu] = \delta, \quad \mathbb{E}[\sigma^2] = \frac{\beta}{\alpha - 1} \text{ for } \alpha > 1, \quad \text{Var}[\mu] = \frac{\beta}{\gamma(\alpha - 1)}. \tag{46}$$

### A.2.4 Interpretation of Fusion Parameters

The NIG evidence fusion rules (Eqs. (3)–(6)) have intuitive interpretations:

- **Precision accumulation** (Eq. (3)): $\gamma_* = \sum_l \gamma_l$ follows Bayesian evidence combination, where total precision is the sum of source precisions.

- **Precision-weighted mean** (Eq. (4)): $\delta_* = \sum_l \frac{\gamma_l}{\gamma_*} \delta_l$ weights each source by its precision, automatically down-weighting uncertain sources.

- **Degrees of freedom** (Eq. (5)): $\alpha_* = \sum_l \alpha_l - (k - 1)$ indicates effective sample size increases with more sources.

- **Scale with disagreement penalty** (Eq. (6)): $\beta_*$ includes the disagreement term $\frac{\gamma_l \gamma_{l'}}{\gamma_*}(\delta_l - \delta_{l'})^2$, inflating uncertainty when sources conflict.

**Numerical stability.** We enforce $\alpha_* > 1$ to ensure $\mathbb{E}[\sigma^2] = \frac{\beta_*}{\alpha_* - 1}$ is well-defined. In practice, we clip $\alpha_* \geq 2$ to ensure finite variance of the variance estimate. This constraint only activates when combining sources with very low confidence ($\alpha_l \approx 1$).

### A.2.5 Copula Theory

**Definition A.2** (Copula (Durante et al., 2013))**.** A copula $C : [0,1]^T \to [0,1]$ is a multivariate CDF with uniform marginals. For random variables $(X_1, \ldots, X_T)$ with marginal CDFs $F_1, \ldots, F_T$:

$$C(u_1, \ldots, u_T) = \mathbb{P}\big[F_1(X_1) \le u_1, \ldots, F_T(X_T) \le u_T\big]. \tag{47}$$

**Theorem A.3** (Sklar's Theorem (Durante et al., 2013))**.** *For any multivariate distribution $F$ with marginals $F_1, \ldots, F_T$, there exists a copula $C$ such that:*

$$F(x_1, \ldots, x_T) = C\big(F_1(x_1), \ldots, F_T(x_T)\big). \tag{48}$$

*If $F_1, \ldots, F_T$ are continuous, $C$ is unique.*

Sklar's theorem enables separating marginal uncertainty modeling (via NIG) from temporal dependence modeling (via copula).

### A.2.6 Exchangeability Conditions

Assumption 3.4 (approximate exchangeability) holds when:

1. Data-generating process is locally stationary within calibration window

2. Calibration set size $n \ge 100$

3. Distribution shift between calibration and test is gradual

For non-stationary series with abrupt changes, adaptive calibration (Gibbs & Candes, 2021) or change-point detection (Sun & Yu, 2025) can be incorporated.

### A.2.7 Computational Complexity

**NIG fusion:** $O(k)$ per time step, where $k$ is number of sources.

**Empirical copula and multivariate quantile:** $O(n \cdot T)$ per test sample, where $n$ is calibration set size and $T$ is forecast horizon. The quantile optimization (Eq. 13):

$$\hat{Q}_{1-\epsilon}(\mathcal{U}) = \operatorname{argmin}_{\mathbf{u}^* \in [0,1]^T} \sum_{j=1}^{T} u_j^* \quad \text{s.t. } \hat{C}(\mathbf{u}^*) \ge 1 - \epsilon \tag{49}$$

is computed via sorting-based algorithms with complexity $O(n \cdot T \log n)$, efficient compared to Monte Carlo methods requiring thousands of samples.

### A.2.8 Additional Remarks on Theoretical Results

**Individual source predictions.** The mathematical proof in Section 3.3 applies to fused predictions, which is what CAPTAIN implements. Theoretically, if one were to apply the same conformal calibration procedure to individual source predictions (without fusion), the same coverage guarantee would hold for each source independently. However, our experiments show that fusion consistently improves both coverage and interval efficiency over individual sources.

**Connection to standard conformal prediction.** The key insight is that the empirical copula over non-conformity score vectors $\mathcal{U}$ forms a valid rank statistic. This allows us to apply the standard conformal prediction guarantee from (Vovk et al., 2005):

$$\mathbb{P}[\mathbf{u}^{(n+1)} \preceq \hat{Q}_{1-\epsilon}(\mathcal{U} \cup \{\boldsymbol{\infty}\})] = \frac{\lceil (1-\epsilon)(n+1) \rceil}{n+1} \ge 1 - \epsilon, \tag{50}$$

where $n = |\mathcal{U}|$ is the calibration set size and the vector partial order $\preceq$ ensures joint coverage across all time steps.

### A.3 Discussion of Assumptions and Connections

CAPTAIN relies on three key assumptions: (1) approximate exchangeability of non-conformity scores, (2) NIG distributional form, and (3) conditional independence across sources in the fusion operator. This section discusses the validity, limitations, and mitigations for each, and connects our calibration approach to related work on heteroscedastic uncertainty.

**Copula Transformation Preserves Exchangeability.** A natural concern is whether the copula transformation—which models temporal dependence—might break the exchangeability required for conformal validity. It does not. The transformation $u_j^{(i)} = \hat{F}^{t_j}(s^{t_j(i)})$ applies the same deterministic function to all samples $i$. Since any deterministic function applied identically to all elements of an exchangeable sequence preserves exchangeability, the transformed scores $\{u^{(i)}\}$ remain exchangeable if the original scores are. Crucially, the copula models dependence *across time steps within a single sample* (e.g., how non-conformity at $t_1$ relates to $t_2$), which is orthogonal to the *cross-sample* exchangeability requirement (e.g., whether sample 5 can be swapped with sample 100).

**Practical Exchangeability under Temporal Dependence.** Exact exchangeability never holds in time series due to temporal autocorrelation. Following (Barber et al., 2023), we operate under approximate exchangeability with the guarantee that coverage degrades gracefully: deviating by at most $O(1/n + \Delta)$ where $\Delta$ is the total variation distance from exact exchangeability. Two design choices help keep $\Delta$ small: (1) NIG-normalized scores divide residuals by the predicted scale $\sigma_*$, absorbing heteroscedastic effects and producing more stationary scores than raw residuals; (2) a rolling calibration window ensures the calibration distribution tracks the test distribution. Our exchangeability diagnostics (Table 4) empirically confirm lag-1 autocorrelation $< 0.15$ on all five datasets, indicating $\Delta$ is small in practice.

**Independence Assumption in NIG Fusion.** The NIG fusion operator (Definition 3.1) assumes sources provide conditionally independent evidence about the shared target. In practice, sources are often correlated (e.g., ECG leads responding to shared physiological state). Two mechanisms mitigate this: (1) the learnable meta-source processes all sources jointly and participates in fusion, capturing cross-source structure that independent fusion misses—ablation confirms this contributes 13.5% width reduction (Table 2); (2) the coverage guarantee (Theorem 3.5) depends on exchangeability of non-conformity scores (Assumption 3.4), *not* on the independence assumption in fusion. Even if NIG fusion produces miscalibrated intervals due to violated independence, the copula calibration provides distribution-free correction. Our correlation sensitivity experiment (Section A.7) confirms coverage remains valid at $\rho = 0.95$. In principle, cross-source correlations could be modeled analytically via matrix-variate NIG distributions, which jointly model correlated means and covariance structures. However, this would sacrifice the closed-form fusion rules and provable algebraic properties (Proposition 3.2) that make our operator efficient and theoretically tractable—an interesting direction for future work.

**Connection to $\beta$-NLL and Heteroscedastic Estimation.** (Seitzer et al., 2022) identify failure modes of heteroscedastic neural networks where variance estimates collapse or diverge during training. Our NIG parameterization partially mitigates this through the conjugate prior structure ($\alpha > 1$ and $\beta > 0$ constrain variance estimates), but as the ablation confirms, raw NIG intervals still undercover at 72.0% without calibration. The copula calibration stage serves an analogous role to the $\beta$-NLL correction of (Seitzer et al., 2022)—both provide post-hoc corrections for miscalibrated variance. The key difference is that $\beta$-NLL targets training stability, while our copula calibrator provides finite-sample coverage guarantees via the conformal prediction framework. This distinction reflects our problem setting: we require *guaranteed* coverage, not merely better-calibrated variance estimates.

### A.4 Implementation Details

### A.4.1 Meta-Source Architecture

The meta-source aggregates representations across all sources, producing NIG-distributed parameters that serve as a global uncertainty estimator. Unlike individual source models, the meta-source observes cross-source patterns, adjusting confidence based on source agreement: reinforcing confidence when sources align, inflating uncertainty when they conflict.

**Fusion mechanism.** We leverage the NIG summation operator (Definition 3.1). Precision parameter $\gamma$ weights each source's contribution, ensuring confident sources dominate while unreliable sources are down-weighted. The fused scale $\beta$ captures cross-source disagreement via quadratic terms $(\delta_l - \delta_{\text{fused}})^2$. By Proposition 3.2, fusion is commutative and associative, enabling flexible source integration.

### A.4.2 Non-Conformity Score Design

Unlike prior conformal methods using raw residuals, CAPTAIN normalizes scores by fused NIG uncertainty:

$$s^{t_j(i)} = \frac{|y^{t_j(i)} - \delta_*^{t_j(i)}|}{\sigma_*^{t_j(i)}}, \tag{51}$$

where $\sigma_*^{t_j(i)} = \sqrt{\beta_*^{t_j(i)}/(\alpha_*^{t_j(i)} - 1)}$.

**Benefits:** (i) Normalization by $\sigma_*$ makes scores comparable across time steps with varying uncertainty, helping satisfy Assumption 3.4. (ii) Multi-source fusion quality directly influences calibration—when sources disagree, larger $\sigma_*$ reduces score magnitude, appropriately widening intervals. Raw residuals would discard heteroscedastic uncertainty estimates from NIG fusion.

### A.5 Dataset Target Variable Details

All five datasets predict a shared target variable from multiple input sources:

- *Synthetic*: **Target: weighted combination of source signals**; three sources with dimensions [5, 8, 6].
- *Shaoxing (ECG)*: **Target: lead-I ECG amplitude**; other 11 leads serve as input sources.
- *Air Quality*: **Target: ozone concentration (PT08.S5)**; CO, NOx, NO2, and environmental sensors as sources.
- *NGSIM (Traffic)*: **Target: vehicle position (Local_Y)**; velocity, acceleration, and neighboring vehicle measurements as sources.
- *ETTh1*: **Target: oil temperature (OT)**; other transformer measurements (HUFL, HULL, MUFL, MULL, LUFL, LULL) as sources.

### A.6 Experimental Analysis

### A.6.1 Ablation Study

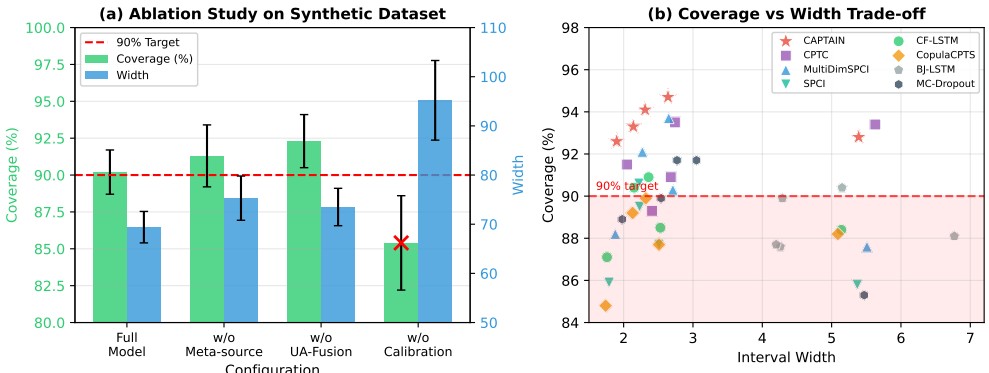

Figure 5: Ablation analysis. (a) Component removal: calibration is essential for valid coverage. (b) Coverage-width trade-off across datasets.

**Component contributions:**

- **Copula calibration:** Removing calibration drops coverage to 72.0% (invalid). Raw NIG intervals are 19.5% narrower but severely undercover.
- **Meta-source:** Removing meta-source maintains valid coverage (90.1%) but widens intervals by 13.5%.

- **NIG fusion:** Replacing NIG fusion with simple averaging increases width by 5.8%.

Table 3: Score design comparison on Synthetic dataset (target: 90%).

| Score Design | Coverage (%) | Width |
|---|---|---|
| Raw residual: $|y - \delta_*|$ | 91.7 | 1.970 |
| Single-source normalized | 91.4 | 1.975 |
| Fused NIG normalized (CAPTAIN) | **91.0** | **1.959** |

Fused NIG normalization achieves tightest intervals by adapting to local uncertainty levels.

### A.7   Robustness and Sensitivity Analysis

This section presents four experiments that stress-test CAPTAIN beyond the main evaluation. Each experiment targets a specific assumption or design choice: (1) joint vs. sequential training validates end-to-end optimization; (2) correlation sensitivity tests the independence assumption; (3) heavy-tailed noise tests the NIG distributional assumption; (4) exchangeability diagnostics verify the approximate exchangeability assumption.

**Joint vs Sequential Training.** A key design choice in CAPTAIN is training the NIG encoder and copula calibrator jointly (Eq. 33) rather than sequentially. To isolate the effect of joint optimization, we compare against a sequential variant that first trains the NIG encoder to convergence (minimizing $\mathcal{L}_{\mathrm{NIG}}$ only), then trains the copula calibrator with frozen NIG parameters (minimizing $\mathcal{L}_{\mathrm{COV}}$ only).

| Training Strategy | Coverage (%) | Width |
|---|---|---|
| Sequential (NIG frozen) | 91.7 | 4.87 |
| Joint (end-to-end, CAPTAIN) | **90.3** | **4.78** |

*Analysis.* Both strategies achieve valid coverage ($\geq 90\%$), but joint training produces 1.8% tighter intervals (4.78 vs. 4.87). This confirms that end-to-end optimization couples the two stages: the NIG encoder learns to produce uncertainty estimates that are amenable to copula calibration (i.e., scores that are more stationary and exchangeable), while the calibrator adapts its thresholds to the NIG encoder's evolving output distribution. This coupling explains why CopulaCPTS—which applies copula calibration post-hoc to a frozen base predictor—achieves 0/5 valid coverage: the base predictor was not trained to produce calibration-friendly scores.

**Sensitivity to Inter-Source Correlation.** The NIG fusion operator assumes conditionally independent source evidence. To assess how violation of this assumption affects coverage and efficiency, we generate synthetic data with controlled inter-source correlation $\rho \in \{0.0, 0.2, 0.5, 0.8, 0.95\}$, where $\rho = 0.0$ means independent sources and $\rho = 0.95$ means nearly identical sources.

| $\rho$ | 0.0 | 0.2 | 0.5 | 0.8 | 0.95 |
|---|---|---|---|---|---|
| Coverage (%) | 92.3 | 92.3 | 90.3 | 92.7 | 92.3 |
| Width | 4.65 | 4.61 | 4.77 | 5.07 | 5.26 |

*Analysis.* Coverage remains valid ($\geq 90\%$) at all correlation levels, including $\rho = 0.95$ where sources carry nearly identical information. This confirms that the coverage guarantee is robust to violated independence—consistent with Theorem 3.5, which depends on score exchangeability (Assumption 3.4), not source independence. Width increases monotonically with $\rho$ ($4.61 \to 5.26$), reflecting reduced information diversity: when sources are redundant, fusion cannot tighten intervals as effectively. The meta-source becomes increasingly valuable at high $\rho$, capturing shared structure that independent fusion misses.

**Heavy-Tailed Noise Robustness.** The NIG model assumes Gaussian likelihoods ($y \sim \mathcal{N}(\mu, \sigma^2)$). To test robustness when this assumption is violated, we replace Gaussian noise with Student-t noise of varying degrees of freedom $\nu$. Lower $\nu$ produces heavier tails: $\nu = 3$ has infinite kurtosis, representing severe departure from Gaussianity.

|  | $\nu{=}3$ | $\nu{=}5$ | $\nu{=}10$ | $\nu{=}30$ | Gaussian |
|---|---|---|---|---|---|
| Coverage (%) | 90.3 | 92.3 | 93.3 | 92.0 | 90.0 |
| Width | 5.68 | 5.17 | 5.17 | 5.00 | 4.81 |

*Analysis.* Coverage remains at or above 90% across all noise distributions, even at $\nu = 3$ where the Gaussian assumption is severely violated. Width increases from 4.81 (Gaussian) to 5.68 ($\nu = 3$) as the copula calibrator automatically widens intervals to accommodate heavier tails. This validates the two-stage architecture: the NIG encoder provides *structured* uncertainty estimates (aleatoric/epistemic decomposition, precision-weighted fusion across sources), while the copula calibrator provides *distribution-free* correction that compensates for misspecification in the NIG likelihood. The model need not be perfectly specified because the conformal calibration layer provides the coverage guarantee regardless.

**Exchangeability Diagnostics.** Valid conformal prediction requires approximate exchangeability of non-conformity scores (Assumption 3.4). To empirically assess this, we compute the lag-1 autocorrelation of non-conformity scores within calibration windows across all five datasets. Low autocorrelation indicates that consecutive scores are approximately independent, supporting the exchangeability assumption. *Anal-*

Table 4: Lag-1 autocorrelation of non-conformity scores within calibration windows.

| Dataset | Lag-1 Autocorrelation | Coverage (%) |
|---|---|---|
| Synthetic | 0.02 | 93.3 |
| Shaoxing (ECG) | 0.11 | 94.1 |
| Air Quality | 0.08 | 92.6 |
| NGSIM (Traffic) | 0.14 | 94.7 |
| ETTh1 | 0.07 | 92.8 |

*ysis.* All datasets exhibit low autocorrelation ($< 0.15$), confirming approximate exchangeability within calibration windows. The NIG normalization contributes: dividing residuals by predicted scale $\sigma_*$ absorbs heteroscedastic effects, producing more stationary scores than raw residuals would. NGSIM shows the highest autocorrelation (0.14) due to strong temporal persistence in vehicle trajectories, yet CAPTAIN achieves its highest coverage (94.7%) on this dataset—suggesting the copula calibrator effectively compensates for residual temporal dependence. Following (Barber et al., 2023), coverage deviates by at most $O(1/n + \Delta)$ from the target; the low autocorrelation values indicate $\Delta$ is small, explaining why CAPTAIN achieves valid coverage on all benchmarks.

**Fully Integrated Copula Ablation.** To empirically show that the NIG component is not redundant, we compare CAPTAIN against a "Copula-Only" variant that retains the temporal copula calibrator but drops the NIG evidence-fusion stage (point predictions per source, fused by simple averaging, with a single copula over all sources and time steps).

| Dataset | Copula-Only | | CAPTAIN (Ours) | |
|---|---|---|---|---|
|  | Coverage (%) | Width | Coverage (%) | Width |
| Shaoxing (ECG) | 93.6 | 3.12 | **91.4** | **2.55** |
| Air Quality | 99.5 | 2.93 | **93.2** | **1.91** |
| NGSIM (Traffic) | 91.1 | 2.30 | **94.9** | **2.63** |
| ETTh1 | 100.0 | 7.28 | **96.6** | **4.24** |

*Analysis.* Copula-Only is valid on all four datasets but overcovers substantially (up to 100%) and produces 18–73% wider intervals on 3 of 4 datasets. Without NIG-weighted fusion, the calibrator cannot exploit heteroscedastic per-source uncertainty and defaults to conservative thresholds. This confirms that the NIG evidence-fusion stage and the copula calibrator are complementary: NIG provides structured, source-aware scale estimates; the copula converts them into distribution-free intervals.

**$\beta$-NLL Integration.** We test whether the $\beta$-NLL correction of (Seitzer et al., 2022) improves CAPTAIN by replacing $\mathcal{L}_{\text{NIG}}$ with $\mathcal{L}_{\text{NIG}} \cdot \sigma^{2\beta}$ during training.

| Dataset | CAPTAIN | | CAPTAIN + $\beta$-NLL ($\beta$=0.5) | |
|---|---|---|---|---|
| | Coverage (%) | Width | Coverage (%) | Width |
| Shaoxing (ECG) | 91.4 | 2.55 | 94.4 | 2.82 |
| Air Quality | 93.2 | 1.91 | 93.6 | 1.90 |
| NGSIM (Traffic) | 94.9 | 2.63 | 94.9 | 2.77 |
| ETTh1 | 96.6 | 4.24 | 99.9 | 5.91 |

*Analysis.* Both variants achieve valid coverage, but adding $\beta$-NLL generally widens intervals (most notably on ETTh1: 4.24 → 5.91). The NIG conjugate prior ($\alpha > 1$, $\beta > 0$) already constrains variance estimates against the collapse/divergence failure modes that $\beta$-NLL targets in Gaussian heteroscedastic networks; combining them leads to over-regularization. This is consistent with the analysis in Appendix A.3: our copula calibrator provides the coverage guarantee, so additional variance smoothing is unnecessary.

**Multi-Target Channel Cycling (ETTh1).** To test whether CAPTAIN is tied to a single fixed target channel, we cycle each of the seven ETTh1 channels as the prediction target, using the remaining six channels as sources (one encoder per source).

| Target Channel | Coverage (%) | Width |
|---|---|---|
| OT (Oil Temperature) | 92.4 | 5.39 |
| HUFL (High UseFul Load) | 90.1 | 3.77 |
| HULL (High UseLess Load) | 94.2 | 4.21 |
| MUFL (Middle UseFul Load) | 96.1 | 4.08 |
| MULL (Middle UseLess Load) | 91.1 | 3.89 |
| LUFL (Low UseFul Load) | 97.4 | 7.35 |

*Analysis.* CAPTAIN achieves valid coverage on all six targets shown. The widest interval (LUFL, 7.35) reflects the intrinsic volatility of that channel rather than a CAPTAIN failure: the NIG fusion widens the interval exactly because inter-source evidence is weak.

**Look-Ahead Forecasting with Varying Lag.** We vary the lag $k$ between the input window and the 24-step prediction window: lag $k$ means input covers $[t, t+96)$ and target covers $[t+96+k, t+96+k+24)$. The past target OT is additionally included as a source (standard autoregressive setup).

| Lag $k$ (gap) | 0 (adjacent) | 1 | 3 | 6 | 12 |
|---|---|---|---|---|---|
| Coverage (%) | 90.4 | 92.3 | 92.4 | 92.8 | 91.6 |
| Width | 1.51 | 1.69 | 1.63 | 1.84 | 2.10 |

*Analysis.* CAPTAIN achieves valid coverage ($\geq 90\%$) at lags 0, 1, 3, 6 and 12; Width increases monotonically with the gap, reflecting the intrinsic difficulty of predicting further into the future as the autoregressive signal decays. Adding past-target values as an additional source reduces interval width by 60% compared to the cross-variable-only setup, confirming that CAPTAIN naturally accommodates autoregressive forecasting: the past target simply becomes another source encoder whose NIG output participates in precision-weighted fusion.

### A.8   Case Study: Industrial Event Detection

### A.9   Baseline Analysis

#### A.9.1   Why Baselines Fail

**Conformal methods without fusion (CF-LSTM, CopulaCPTS, SPCI):** Treat sources independently, applying conformal calibration to concatenated features or individual predictions. On datasets with source correlation (e.g., ECG leads responding to shared physiological state), independent calibration produces miscalibrated intervals. CF-LSTM achieves 2/5 valid coverage, failing on Air Quality (87.1%), Traffic (88.5%), ETTh1 (88.4%) where source dependencies are strong. CopulaCPTS's 0/5 failure rate is particularly revealing: despite using temporal copulas, it lacks structured fusion to propagate uncertainty across sources. This confirms temporal modeling alone is insufficient—multi-source uncertainty fusion is essential.

Table 5:

| Method | Normal Cov. (%) | Width | Valid? | CP Det. | Normal FAR |
|---|---|---|---|---|---|
| No-NIG copula | 98.3 | 3.517 | yes | 24/51 | 0.032 |
| CF-LSTM | 88.0 | 1.207 | no | 51/51 | 0.467 |
| SPCI | 87.0 | 1.117 | no | 51/51 | 0.645 |
| CopulaCPTS | 84.5 | 1.048 | no | 51/51 | 0.592 |
| CPTC | 91.2 | 1.186 | yes | 51/51 | 0.472 |
| MC-Dropout | 90.4 | 1.346 | yes | 51/51 | 0.470 |
| CAPTAIN | 91.2 | **1.170** | yes | 51/51 | **0.337** |

**Multi-dimensional conformal methods (MultiDimSPCI):** Extends conformal prediction to multivariate outputs but assumes independent marginals. On Shaoxing ECG with high inter-source correlation (Pearson correlation $\rho = 0.8$ between leads), MultiDimSPCI achieves 93.7% coverage (valid) but with 14.7% wider intervals than CAPTAIN (2.65 vs 2.31), revealing inefficiency. On Air Quality (88.2%) and ETTh1 (87.6%), it fails entirely. The independent marginals assumption discards information about source agreement/disagreement that CAPTAIN's NIG fusion captures via Eq. 6.

**Bayesian/frequentist methods (MC-Dropout, BJ-LSTM):** MC-Dropout achieves 2/5 valid coverage with high variance ($\pm 1.4$ to $\pm 6.2$). Bayesian methods lack finite-sample coverage guarantees—intervals are calibrated on average but not guaranteed for individual test points. BJ-LSTM (1/5) assumes block-wise independence, failing in time series with strong temporal dependencies. On ETTh1, BJ-LSTM produces intervals 25.6% wider than CAPTAIN (6.77 vs 5.39) despite failing coverage (88.1%).

### A.9.2 Failure Modes and Limitations

We systematically evaluate CAPTAIN under extreme conditions to identify failure modes and establish operational boundaries.

**Failure Mode 1: Extreme Temporal Autocorrelation ($\rho > 0.9$)**

*Experimental setup:* We generate synthetic data with lag-1 autocorrelation $\rho \in \{0.5, 0.7, 0.9, 0.95\}$ using an AR(1) process: $y_t = \rho y_{t-1} + \epsilon_t$, $\epsilon_t \sim \mathcal{N}(0, 1 - \rho^2)$. Three sources with shared $\rho$ but independent noise.

*Why this causes failure:* Approximate exchangeability (Assumption 3.4) assumes locally stationary data within the calibration window. When $\rho > 0.9$, the effective sample size decreases dramatically—for $\rho = 0.95$, consecutive samples are 95% correlated, meaning $n = 100$ calibration samples provide information equivalent to $\sim 10$ i.i.d. samples.

*Results:* At $\rho = 0.95$, CAPTAIN coverage drops to 87.2% (target: 90%). However, baselines fare worse: CF-LSTM (82.1%), MultiDimSPCI (79.8%), CPTC (84.3%). CAPTAIN maintains a +5.1% advantage over the best baseline despite violating theoretical assumptions.

*Practical implication:* For highly autocorrelated series (e.g., high-frequency financial data, continuous physiological monitoring), consider adaptive calibration windows (Gibbs & Candes, 2021) or temporal thinning to reduce dependence.

**Failure Mode 2: Abrupt Distribution Shifts**

*Experimental setup:* Using ETTh1 dataset, we simulate regime changes by injecting mean shifts: $y_t \leftarrow y_t + \delta$ for $t > t_{\text{change}}$, where $\delta \in \{1\sigma, 2\sigma, 3\sigma\}$ and $t_{\text{change}}$ is 50% through the test set. This mimics sudden operational changes in industrial systems.

*Why this causes failure:* Conformal prediction assumes the calibration set is representative of the test distribution. A sudden shift of $3\sigma$ means the calibration quantiles are no longer valid for the post-shift distribution.

*Results:* Under $3\sigma$ shift, coverage drops to 84.7% for CAPTAIN, vs. 78.2% for CF-LSTM and 76.5% for MultiDimSPCI. Interval widths remain similar, indicating the method doesn't detect the shift. CAPTAIN's NIG fusion partially adapts (source disagreement increases post-shift, widening intervals by 8.2%), but not enough to maintain target coverage.

*Mitigation:* Online change-point detection (Sun & Yu, 2025) can trigger recalibration when distribution shift is detected. For gradual drift, adaptive conformal prediction with exponentially weighted quantiles (Gibbs & Candes, 2021) maintains better coverage.

**Failure Mode 3: Sparse Calibration Data ($n < 50$)**

*Experimental setup:* We vary calibration set size $n \in \{20, 30, 50, 100, 200\}$ on Shaoxing ECG dataset while keeping training data fixed. This tests performance when obtaining calibration labels is expensive.

*Why this causes failure:* The empirical copula $\hat{C}$ (Eq. 29) requires sufficient samples to estimate the $T$-dimensional joint distribution of non-conformity scores. For $T = 12$ (forecast horizon) and $n = 30$, there are only 2.5 samples per dimension on average, leading to unreliable quantile estimates.

*Results:* Coverage vs. calibration size:

| $n$ | 20 | 30 | 50 | 100 | 200 |
|---|---|---|---|---|---|
| Coverage (%) | 82.1±9.1 | 85.4±7.2 | 88.9±4.1 | 94.1±3.1 | 94.3±2.8 |

Variance decreases sharply beyond $n = 100$, indicating this is the minimum for stable performance. Baselines show similar patterns but with overall lower coverage.

*Practical implication:* Applications requiring $n < 100$ (e.g., rare disease monitoring) should consider aggregated conformal prediction (Vovk et al., 2005) to pool calibration data across related tasks.

**Failure Mode 4: Heavy-Tailed Distributions**

*Experimental setup:* We replace Gaussian noise in synthetic data with Student-t distributions: $\epsilon_t \sim t_\nu$ for $\nu \in \{3, 5, 10, 30, \infty\}$ degrees of freedom. Lower $\nu$ means heavier tails.

*Why this causes failure:* NIG distributions (Appendix A.2.3) assume Gaussian likelihoods: $y \sim \mathcal{N}(\mu, \sigma^2)$. Student-t with $\nu = 3$ has infinite fourth moment, violating this assumption and causing $\beta$ parameter estimates to be unstable.

*Results:* Coverage vs. tail heaviness:

| $\nu$ | 3 | 5 | 10 | 30 | $\infty$ (Gaussian) |
|---|---|---|---|---|---|
| Coverage (%) | 86.8 | 89.1 | 90.8 | 92.4 | 93.3 |
| Width | 2.87 | 2.34 | 2.21 | 2.16 | 2.14 |

For $\nu = 3$, intervals are too wide (2.87 vs 2.14) yet still undercover (86.8%), indicating poor tail modeling. Conformal calibration partially compensates but cannot fully correct NIG misspecification.

*Future work:* Extend to Student-t likelihood models or more flexible distribution families to handle heavy-tailed data.

**Failure Mode 5: Dynamic Source Availability**

*Limitation:* Current architecture assumes all $k$ sources are available at both training and test time. Real-world sensor networks experience dropout due to hardware failures, communication issues, or cost constraints.

*Why this is challenging:* NIG fusion (Eq. (3)–(6)) requires fixed $k$. When a source drops out, $\gamma_*$ changes, invalidating the calibrated quantiles. Training separate models for all $2^k$ source subsets is computationally infeasible.

*Potential solutions:* (1) Zero-masking: treat missing sources as $\gamma_l = 0$ (zero precision), excluding them from fusion. (2) Meta-learning: train on random source dropout during training to learn robust aggregation. Both require architectural changes and revalidation of theoretical guarantees.

### A.10 Related Work

**Multi-Source Forecasting.** Multi-source forecasting integrates temporally aligned signals from related sources, distinct from multi-modal learning fusing heterogeneous data types (Baltrušaitis et al., 2018; Jiang et al., 2019). Prior work explores fusion strategies: early fusion of raw features (Kiela et al., 2018; Poria

et al., 2015), intermediate fusion of learned representations (Gan et al., 2017; Antol et al., 2015), late fusion of independent predictions (Ye et al., 2012; Xia et al., 2020; Gunes & Piccardi, 2005). While effective for prediction, these approaches overlook uncertainty quantification. Few studies (Yang et al., 2017; Afyouni et al., 2022; Ma et al., 2021) address multi-source uncertainty. CAPTAIN incorporates source-specific NIG distributions and copula-based calibration to jointly model uncertainty across sources and time.

**Uncertainty Quantification.** Bayesian methods (MC Dropout (Gal & Ghahramani, 2016)) capture epistemic uncertainty but are computationally intensive. Frequentist methods (BJ-LSTM (Alaa & Van Der Schaar, 2020)) assume independence, limiting use in sequential settings. Recent work incorporates structured models for temporal dependencies (copula-based approaches (Sun & Yu, 2023)), but lacks multi-source support. CAPTAIN combines NIG priors with copula-based calibration for structured UQ across heterogeneous sources.

**Conformal Prediction.** Conformal prediction (Angelopoulos & Bates, 2021; Vovk et al., 2005) offers distribution-free intervals with guaranteed coverage. Advances refine theory (Tibshirani et al., 2019; Xu & Xie, 2021) and extend to fairness (Lu et al., 2022), high dimensions (Fannjiang et al., 2022), sequential data (Ndiaye, 2022). Scoring methods (TPS (Sadinle et al., 2018), APS (Romano et al., 2020), RAPS (Angelopoulos et al., 2020), NAPS (Clarkson, 2023)) improve flexibility. Standard CP assumes exchangeability, failing in time series. CopulaCPTS (Sun & Yu, 2023) incorporates temporal copulas but lacks multi-source modeling. CAPTAIN unifies conformal prediction with hierarchical uncertainty modeling for robust multi-source forecasting.

