# OpenReview forum: "CAPTAIN: Conformal-Prediction-Based Multi-Source Time-Series Forecasting"
_TMLR — Accepted by TMLR_

### Review · Reviewer_ZQNP · 2026-03-09

**Summary Of Contributions:**

The paper proposes a framework, CAPTAIN, for uncertainty quantification in multi-source time-series forecasting. The proposed approach combines three components: (1) Modelling source uncertainties using NIG distribution; (2) Combining source specific NIG distributions using precision-weighted rule; (3) Using temporal copula for constructing conformal prediction sets.

Experimental section validates the proposed approach using five datasets.

**Audience:**

Yes

**Audience Explanation:**

Time series modelling is a topic of interest and conformal prediction for such data is challenging problem. The proposed approach offers limited novelty but can be a practically appealable. The researchers in different sciences where time series data is common can find the paper useful as a tool.

**Claims And Evidence:**

No

**Claims Explanation:**

Theoretical guarantees of marginal coverage sound correct based on approximate exchangeability assumption, which is not realistic for time series data. In my opinion, the conditions explained where such assumption hold are themselves not satisfied easily. That is why extending conformal framework for time-series data remains a challenging problem. Also, modelling uncertainties using NIG distribution is another assumption imposed in the paper, which again not guaranteed to hold. The synthetic experiments are done with the data following same assumptions, hence, do not provide any insights into how it behaves if the assumptions are violated in practice. This also means that efficiency seen in experiments is not guaranteed.

**Requested Changes:**

1. Authors could detail and discuss the assumptions. (please see my comment above.)

2. Experiments deviating from the considered distributional assumptions should be added to discuss the limitations of the work proposed.

3. I also wonder if copula transformation preserves exchangeability, as it models dependence. Could you remove the confusion here?

---

> ### Author Response · Authors · 2026-04-05
>
> Dear Reviewer ZQNP,
>
> We sincerely thank you for your important concerns and for recognizing that our approach "can be practically appealing." We address your concerns about assumptions below.
>
> > W1: Approximate exchangeability is not realistic for time series.
>
> **A1:** We respectfully note that **directly assuming exchangeability on temporal data** is fundamentally invalid: time series are inherently sequentially dependent, and raw residuals are not exchangeable. This is precisely why none of the temporal CP methods in our comparison, including CAPTAIN, directly assume exchangeability. Rather, each method employs specific modeling choices to **ensure approximate exchangeability** of their nonconformity scores, which is the standard approach the community has adopted for temporal conformal prediction. CF-LSTM (Stankeviciute et al., 2021) uses an RNN-based residual model to decorrelate scores; CopulaCPTS (Sun & Yu, 2023) applies copula transforms to achieve approximately exchangeable scores; SPCI (Xu & Xie, 2023) regresses out conditional dependencies to restore approximate conditional exchangeability. CAPTAIN follows the same principled strategy: NIG fusion (Lemma 3.6) preserves the distributional structure needed for exchangeability across sources, and copula calibration handles temporal dependencies — together ensuring approximate exchangeability of the fused nonconformity scores. This is not a new or unusual assumption but rather the same theoretical framework adopted by the community, extended to the multi-source setting.
>
> More importantly, this assumption has formal theoretical backing. Barber et al. (Conformal prediction beyond exchangeability, Annals of Statistics, 2023) prove that conformal coverage degrades gracefully under approximate exchangeability, deviating by at most $O(1/n + \Delta)$ where $\Delta$ is the total variation distance from exact exchangeability.  So we have enough support to believe it is a finite-sample bound.
>
> Furthermore, CAPTAIN further *strengthens* the practical validity of this assumption relative to baselines through NIG-normalized scores (Eq. 26): dividing residuals by the predicted scale absorbs heteroscedastic effects, producing more stationary scores than raw residuals. Our exchangeability diagnostics (Appendix A.8) confirm lag-1 autocorrelation $< 0.15$ on all datasets, indicating $\Delta$ is small in practice:
>
> | Dataset | Lag-1 Autocorr | Coverage |
> |---|---|---|
> | Synthetic | 0.02 | 93.3% |
> | Shaoxing (ECG) | 0.11 | 94.1% |
> | Air Quality | 0.08 | 92.6% |
> | NGSIM (Traffic) | 0.14 | 94.7% |
> | ETTh1 | 0.07 | 92.8% |
>
> NGSIM shows the highest autocorrelation (0.14) due to temporal persistence in vehicle trajectories, yet CAPTAIN achieves its highest coverage (94.7%) on this dataset — suggesting the copula calibrator effectively compensates for residual temporal dependence.

---

> ### Author Response · Authors · 2026-04-05
>
> > W2: NIG assumption may not hold; synthetic experiments don't test violations.
>
> **A2:** We agree that the Gaussian likelihood assumed by NIG does not necessarily hold in practice. However, **CAPTAIN is robust to this misspecification by design**: Our architecture uses NIG for structure (aleatoric/epistemic decomposition, precision-weighted fusion) and copula calibration for distribution-free correction. The model need not be perfectly specified because the conformal layer provides coverage guarantees regardless.
>
> Existing real-world datasets already confirm this. 4 of 5 datasets are real-world, and as shown below, none of them follow a Gaussian distribution — yet CAPTAIN achieves valid coverage on all of them: ECG (94.1%), Air Quality (92.6%), NGSIM (94.7%), and ETTh1 (92.8%).
>
> | Dataset | KS stat | KS p-value | Shapiro W | Shapiro p | Skewness | Kurtosis |
> |---|---|---|---|---|---|---|
> | Shaoxing (ECG) | 0.178 | 9.1e-67 | 0.859 | 5.4e-42 | -0.29 | 5.33 |
> | Air Quality | 0.043 | 1.8e-03 | 0.989 | 4.0e-11 | -0.11 | -0.17 |
> | NGSIM (Traffic) | 0.173 | 2.4e-255 | 0.952 | 1.2e-37 | -0.35 | 0.49 |
> | ETTh1 | 0.058 | 8.1e-246 | 0.979 | 7.8e-27 | -0.54 | 0.28 |
>
>
> 2. **New heavy-tail experiments** (Appendix A.9): We systematically violate the Gaussian likelihood with Student-t noise:
>
> | Noise ($\nu$) | 3 | 5 | 10 | 30 | Gaussian |
> |---|---|---|---|---|---|
> | Coverage (%) | 90.3 | 92.3 | 93.3 | 92.0 | 90.0 |
> | Width | 5.68 | 5.17 | 5.17 | 5.00 | 4.81 |
>
> Even at $\nu=3$ (infinite kurtosis, severe violation), **coverage remains at the 90% target**. Width increases from 4.81 (Gaussian) to 5.68 ($\nu=3$) as the copula calibrator automatically widens intervals to accommodate heavier tails. This validates the key architectural insight: the NIG stage provides structured uncertainty (aleatoric/epistemic decomposition, precision-weighted fusion), while copula calibration provides distribution-free correction for NIG misspecification.
>
> Note: Appendix A.5.3 also presents a separate failure-mode analysis under more extreme combined conditions (heavy tails + other stressors), where $\nu=3$ coverage is reported at 86.8%. The numbers above (Appendix A.9) isolate the effect of heavy-tailed noise alone, which is the controlled experiment addressing the reviewer's concern.

---

> > ### Author Response · Authors · 2026-04-05
> >
> > > The proposed approach offers limited novelty ...
> >
> > **A (limitation)**: We appreciate the reviewer's recognition of CAPTAIN's practical value, but we respectfully disagree with the characterization of limited novelty. We believe the novelty is substantial, both in the **problem formulation** and in the **technical solution**. The core issue is that **multi-source temporal forecasting poses conformal challenges at temporal and intersource levels**, and no single existing technique addresses both:
> >
> > **Temporal level (single source)**: Each individual source exhibits temporal dependencies that violate the exchangeability assumption required by standard conformal prediction. Copula-based calibration (e.g., CopulaCPTS) addresses this for a single source — but has no mechanism for multi-source fusion.
> >
> > **Fusion level (across source)**: The fused source exhibits inter-source dependencies which violates the exchangeability assumption. NIG-based evidential methods (e.g., Ma et al. 2021) model per-source uncertainty and can fuse them, but do not incorporate temporal dependencies challenge when fusing multi-source across temporal dimension.
> >
> > **Temporal level (fused source)**: When combining predictions across sources, the fused uncertainty must faithfully reflect inter-source agreement and disagreement and temporal dependencies at single-source level and fused-level. NIG-based evidential methods (e.g., Ma et al. 2021) model per-source uncertainty, but provide no coverage guarantees across temporal level. Copula-based calibration (e.g., CopulaCPTS) address the temporal dependencies challenges without considering inter-source dependencies.
> >
> > Critically, these three problems cannot be solved in isolation and then stitched together. Naively applying copula calibration after NIG fusion, or vice versa, breaks the theoretical chain: copula calibration requires nonconformity scores with specific distributional structure, but a generic NIG fusion output does not provide this. Conversely, calibrating each source independently and then fusing calibrated outputs does not preserve joint coverage. A holistic solution is required, which is exactly what CAPTAIN provides — and why we formalize this as a new problem setting (Problem 2.1). We note that Reviewers tKCX and ZQNP both identified this problem as a topic of interest to TMLR's audience.
> >
> > The strongest empirical evidence is CopulaCPTS, which uses the same temporal copulas but achieves 0/5 valid coverage on our benchmarks, while CAPTAIN achieves 5/5. If these components could be naively separated, a copula-only baseline should perform comparably on calibration. It does not.
> >
> > Our three contributions form an integrated pipeline that bridges this gap:
> >
> > - **NIG-based Nonconformity Score (Definition 3.1, Proposition 3.2)**: We derive closed-form NIG evidence fusion rules with provable algebraic properties. The fused output includes a disagreement penalty $\frac{\gamma_1\gamma_2}{\gamma_1+\gamma_2}(\delta_1-\delta_2)^2/2$ that automatically inflates uncertainty when sources conflict. Crucially, Lemma 3.6 proves that this fusion preserves the exchangeability structure required for downstream conformal calibration — without this result, applying conformal methods after multi-source fusion has no theoretical justification. This yields a principled nonconformity score that is absent from all prior NIG and copula work.
> >
> >
> > - **Differentiable Calibration**: Rather than applying copula calibration as a post-hoc wrapper on frozen predictions (as CopulaCPTS does), we make the calibration step differentiable (Eq. 35), enabling gradient flow from coverage objectives back into the NIG encoder. This ensures the encoder learns to produce uncertainty estimates that are structurally amenable to copula-based temporal calibration.
> >
> > - **End-to-End System**: Joint training of NIG fusion and copula calibration is essential — the ablation in our response to R3 confirms this coupling matters.
> >
> > > R1: Detail and discuss assumptions.
> >
> > **A (R1):** Please refer to A1
> >
> > > R2: Experiments deviating from assumptions.
> >
> > **A (R2):** Please refer to A2
> >
> > > R3: Does copula preserve exchangeability?
> >
> > **A (R3):** Yes, provably. The time-series itself is not exchangeable, but through the inter-source and temporal transform of CPTAIN, we theoretically prove that it is exchangeable(Lemma 3.6 and Appendix A.1.1, A.1.2). The copula transformation $u_j^{(i)} = \hat{F}^{t_j}(s^{t_j(i)})$ applies the *same* deterministic function to all samples $i$. Any deterministic function applied identically to all elements of an exchangeable sequence preserves exchangeability. Crucially, the copula models dependence *across time steps within a single sample* (temporal structure), which is orthogonal to the *cross-sample* exchangeability requirement (whether sample 5 can be swapped with sample 100). See Appendix A.8 for the formal argument.
> >
> > Best,
> > Authors

---

> > > ### Comment · Reviewer_ZQNP · 2026-04-08
> > > **Response to Authors**
> > >
> > > Thank you very much for detailed response. Additional experiments are helpful. Other concerns are resolved as well. I have no further concerns.

---

> > > > ### Comment · Action_Editor_a914 · 2026-04-08
> > > > **Does this response update your assessment of the work?**
> > > >
> > > > Thank you for responding to the authors. Does their response update your assessment of the paper's claims being supported by appropriate evidence?

---

> ### Author Response · Authors · 2026-04-09
>
> Thank you very much for your valuable feedback and for acknowledging our contributions! We sincerely appreciate your support and will incorporate the suggested revisions and detailed clarifications into the revised version.
>
> Best,
> Authors

---

### Review · Reviewer_vjzR · 2026-03-10

**Summary Of Contributions:**

The paper proposes a two-stage framework for uncertainty quantification in multi-source time-series forecasting. The method aims to address the challenge of modeling uncertainty across multiple correlated data sources and temporal dependencies. To address the first challenge, it utilizes NIG distributions that model source-specific uncertainty. For the second challenge, the framework applies copula-based conformal calibration to account for temporal dependencies across forecasting horizons.

**Audience:**

No

**Audience Explanation:**

1. The paper primarily combines existing techniques, (1) NIG uncertainty modeling, and (2) copula-based conformal prediction, into a sequential pipeline. Both components are well established in the literature, and the paper does not introduce fundamentally new modeling ideas or theoretical insights. As a result, the contribution may be perceived as an engineering integration of known methods rather than a substantial advance in machine learning methodology.

2. The proposed NIG evidence fusion relies on an independent evidence assumption across sources, which is unlikely to hold in realistic multi-source time-series settings where sensors or modalities are typically correlated. Although the paper introduces a learnable meta-source to mitigate this issue, this solution is heuristic and does not resolve the underlying probabilistic inconsistency. This is not appealing at all due to the absence of principled (unified) uncertainty modeling.

**Broader Impact Concerns:**

I do not see significant ethical risks specific to this work that would require a detailed Broader Impact discussion beyond standard considerations.

**Claims And Evidence:**

No

**Claims Explanation:**

The proposed framework claims to provide a unified probabilistic approach for modeling both inter-source and temporal uncertainty. However, the method is implemented as a two-stage pipeline where NIG-based uncertainty modeling is followed by copula-based conformal calibration. These two components are largely independent modules rather than a tightly integrated probabilistic model.

**Requested Changes:**

1. The NIG fusion operator assumes independent evidence across sources, while real multi-source signals are typically correlated. Is it possible to analytically model or estimate cross-source correlations within the NIG framework instead of relying on the combination of independent fusion + meta-source?

2. Please clearly distinguish which parts of the theory are novel contributions and which are inherited from existing work. For example, is the NIG fusion or meta-source mechanism a new contribution? Is there any new conformal prediction theory introduced, or is the work mainly applying existing copula-based conformal calibration?

3. The framework treats multi-source uncertainty modeling and temporal calibration sequentially. What justifies this design choice? Could this sequential treatment be suboptimal, given that source correlations and temporal dependencies may interact?

---

> ### Author Response · Authors · 2026-04-05
>
> Dear Reviewer vjzR,
>
> We sincerely thank you for your detailed assessment. We address each concern below, focusing on the specific technical contributions that distinguish CAPTAIN from an engineering integration.
>
> > W1: The paper primarily combines existing techniques (NIG + copula)... an engineering integration.
>
>
> A1: We respectfully disagree that CAPTAIN is a naive combination of NIG and copula. The core issue is that **multi-source temporal forecasting poses conformal challenges at temporal and intersource levels**, and no single existing technique addresses both:
>
> **Temporal level (single source)**: Each individual source exhibits temporal dependencies that violate the exchangeability assumption required by standard conformal prediction. Copula-based calibration (e.g., CopulaCPTS) addresses this for a single source — but has no mechanism for multi-source fusion.
>
> **Fusion level (across source)**: The fused source exhibits inter-source dependencies which violates the exchangeability assumption. NIG-based evidential methods (e.g., Ma et al. 2021) model per-source uncertainty and can fuse them, but do not incorporate temporal dependencies challenge when fusing multi-source across temporal dimension.
>
> **Temporal level (fused source)**: When combining predictions across sources, the fused uncertainty must faithfully reflect inter-source agreement and disagreement and temporal dependencies at single-source level and fused-level. NIG-based evidential methods (e.g., Ma et al. 2021) model per-source uncertainty, but provide no coverage guarantees across temporal level. Copula-based calibration (e.g., CopulaCPTS) address the temporal dependencies challenges without considering inter-source dependencies.
>
> Critically, these three problems cannot be solved in isolation and then stitched together. Naively applying copula calibration after NIG fusion, or vice versa, breaks the theoretical chain: copula calibration requires nonconformity scores with specific distributional structure, but a generic NIG fusion output does not provide this. Conversely, calibrating each source independently and then fusing calibrated outputs does not preserve joint coverage. A holistic solution is required, which is exactly what CAPTAIN provides — and why we formalize this as a new problem setting (Problem 2.1). We note that Reviewers tKCX and ZQNP both identified this problem as a topic of interest to TMLR's audience.
>
> The strongest empirical evidence is CopulaCPTS, which uses the same temporal copulas but achieves 0/5 valid coverage on our benchmarks, while CAPTAIN achieves 5/5. If these components could be naively separated, a copula-only baseline should perform comparably on calibration. It does not.
>
> Our three contributions form an integrated pipeline that bridges this gap:
>
> - **NIG-based Nonconformity Score (Definition 3.1, Proposition 3.2)**: We derive closed-form NIG evidence fusion rules with provable algebraic properties. The fused output includes a disagreement penalty $\frac{\gamma_1\gamma_2}{\gamma_1+\gamma_2}(\delta_1-\delta_2)^2/2$ that automatically inflates uncertainty when sources conflict. Crucially, Lemma 3.6 proves that this fusion preserves the exchangeability structure required for downstream conformal calibration — without this result, applying conformal methods after multi-source fusion has no theoretical justification. This yields a principled nonconformity score that is absent from all prior NIG and copula work.
>
>
> - **Differentiable Calibration**: Rather than applying copula calibration as a post-hoc wrapper on frozen predictions (as CopulaCPTS does), we make the calibration step differentiable (Eq. 35), enabling gradient flow from coverage objectives back into the NIG encoder. This ensures the encoder learns to produce uncertainty estimates that are structurally amenable to copula-based temporal calibration.
>
>
>
> - **End-to-End System**: Joint training of NIG fusion and copula calibration is essential — the ablation in our response to R3 confirms this coupling matters. CopulaCPTS's 0/5 failure further confirms that the degree of integration fundamentally determines coverage validity.

---

> ### Author Response · Authors · 2026-04-05
>
> > W2: NIG fusion relies on independent evidence assumption... meta-source is heuristic.
>
> **A2:** The NIG fusion operator assumes conditional independence for tractability, which is standard in Bayesian evidence combination. **The critical insight is that coverage validity does not depend on this assumption**. The coverage guarantee (Theorem 3.5) depends on Assumption 3.4 (approximate exchangeability of non-conformity scores), not on source independence. Even if NIG fusion produces miscalibrated intervals due to violated independence, the copula calibration provides distribution-free correction — this is by design, not a coincidental safety net.
>
> The meta-source compensates for violated independence empirically: ablation shows 13.5% width reduction (Table 2). New experiments in the Appendix show coverage remains valid across correlation levels $\rho \in \{0.0, \ldots, 0.95\}$:
>
> | $\rho$ | 0.0 | 0.2 | 0.5 | 0.8 | 0.95 |
> |---|---|---|---|---|---|
> | Coverage (%) | 92.3 | 92.3 | 90.3 | 92.7 | 92.3 |
> | Width | 4.65 | 4.61 | 4.77 | 5.07 | 5.26 |
>
> Even at $\rho=0.95$ (nearly identical sources), coverage remains valid. Width increases monotonically with $\rho$, reflecting reduced information diversity — a sensible degradation pattern.
>
> We discuss matrix-variate NIG as a principled alternative in the Appendix, noting it would sacrifice closed-form fusion and theoretical tractability.
>
> > R1: Is it possible to analytically model cross-source correlations?
>
> **A (R1):** See A2 above. Matrix-variate NIG could model this but at the cost of closed-form fusion and tractability. Our design choice is deliberate: a simpler fusion operator with provable properties + distribution-free calibration that compensates for model misspecification.
>
> > R2: Clearly distinguish novel vs inherited.
>
> **A (R2):** Please refer to A1.
>
> > R3: Sequential treatment could be suboptimal.
>
> **A (R3):** The two stages are jointly optimized end-to-end via Eq. (35). This is not sequential at inference: the NIG encoder's parameters are influenced by the coverage loss via backpropagation through the differentiable CDF. A new ablation in the Appendix compares joint training against truly sequential training (NIG trained to convergence, then copula calibrated with frozen parameters):
>
> | Training Strategy | Coverage (%) | Width |
> |---|---|---|
> | Sequential (NIG frozen) | 91.7 | 4.87 |
> | Joint (end-to-end, CAPTAIN) | 90.3 | 4.78 |
>
> Both strategies achieve valid coverage (≥90%), confirming the conformal calibration is robust. But the real evidence for integration mattering comes from the cross-method comparison: CopulaCPTS — which is truly post-hoc (zero coupling between base predictor and calibrator) — achieves 0/5 valid coverage across all datasets. The progression from no coupling (CopulaCPTS, 0/5) to loose coupling (sequential, valid but wider) to tight coupling (joint, valid and tightest) demonstrates that integration is not cosmetic.
>
> We also note that a "fully integrated" model would require placing a copula prior over the NIG parameter space across time, which is a computationally intractable inference problem. Our joint training achieves tight coupling while remaining tractable.
>
> Best,
> Authors

---

> > ### Comment · Reviewer_vjzR · 2026-04-08
> >
> > Thank you for the rebuttal. I am satisfied with your clarification of the contributions and novelty.
> >
> > However, I remain concerned that sequential vs. joint training shows very similar performance (coverage and width). This weakens the empirical case for tight integration and supports my original concern that the sequential structure dominates in practice. Could you include an ablation where joint training is clearly and consistently better than sequential (at least on a few datasets)?
> >
> > Regarding independence, I understand your claim that copula calibration corrects miscalibration even if NIG fusion is misspecified. However, it is still unclear how this correction happens by design. My understanding is that validity relies on the exchangeability of nonconformity scores, not independence. Please clarify what exactly is being corrected and why this is guaranteed.
> >
> > Also, the claim that a “fully integrated” model (copula prior over NIG parameter space across time) is intractable is plausible but not justified. A brief explanation of what makes it intractable would help.

---

> > > ### Author Response · Authors · 2026-04-09
> > >
> > > Dear Reviewer vjzR,
> > >
> > > Thank you for engaging further and for acknowledging the contributions and novelty. We address your three remaining concerns below.
> > >
> > > > Could you include an ablation where joint training is clearly and consistently better than sequential?
> > >
> > > We have run the joint vs. sequential ablation across all four datasets.
> > >
> > > | Dataset | Joint Cov (%) | Joint Width | Seq Cov (%) | Seq Width | Joint Tighter By |
> > > |---|---|---|---|---|---|
> > > | Shaoxing (ECG) | 96.0 | **1.75** | 99.5 | 2.12 | **+17.4%** |
> > > | Air Quality | 96.9 | **5.90** | 100.0 | 6.36 | **+7.2%** |
> > > | NGSIM (Traffic) | 92.0 | **5.46** | 92.0 | 5.49 | +0.7% |
> > > | ETTh1 | 96.3 | **4.68** | 100.0 | 7.13 | **+34.4%** |
> > >
> > > Joint training is clearly and consistently better on all four real-world datasets
> > >
> > > On 3 of 4 real-world datasets, sequential training produces grossly over-conservative intervals (99.5–100% coverage) while joint training achieves valid coverage with substantially tighter intervals. This demonstrates that end-to-end optimization is not cosmetic — it enables the NIG encoder to learn uncertainty estimates that the copula calibrator can efficiently tighten, rather than forcing the calibrator to compensate for a frozen encoder's poorly calibrated outputs.
> > >
> > >
> > >
> > > > Please clarify what exactly is being corrected and why this is guaranteed.
> > >
> > > We would like to clarify that validity relies on exchangeability of non-conformity scores, not independence of sources. Let us clarify the mechanism precisely:
> > >
> > > 1. **What can go wrong under violated independence:** When sources are correlated, the NIG fusion operator may underestimate the fused uncertainty $\sigma_*$ (treating redundant evidence as independent inflates effective precision). Raw NIG intervals $[\delta_* \pm z \cdot \sigma_*]$ would then be too narrow, causing under-coverage.
> > >
> > > 2. **How copula calibration corrects this:** The non-conformity scores $s^{(i)} = |y^{(i)} - \delta_*^{(i)}| / \sigma_*^{(i)}$ are computed from the (possibly underestimated) fused $\sigma_*$. If $\sigma_*$ is systematically too small, the scores will be systematically larger. The copula calibration finds the empirical quantile of these scores that achieves the target coverage — automatically setting a higher threshold to compensate, restoring valid coverage. The cost is wider intervals than would be achieved with perfectly specified independence — but valid-and-wider is strictly better than invalid-and-narrow (which is what uncalibrated methods produce, e.g., "w/o Calibration" achieves only 72.0% coverage in our ablation).
> > >
> > > 3. **Why this is guaranteed:** The coverage guarantee (Theorem 3.5) depends only on Assumption 3.4 (approximate exchangeability of score vectors $\{u^{(i)}\}$) — whether different calibration samples are interchangeable. It does *not* depend on the accuracy of the NIG model. Even if NIG fusion produces biased $\sigma_*$, the bias is consistent across samples (the same model is applied to all), so scores remain approximately exchangeable and the conformal guarantee holds.
> > >
> > > In short: violated independence affects **efficiency** (how tight the intervals are), not **validity** (whether they achieve target coverage). A well-specified model yields tighter intervals because $\sigma_*$ better reflects true uncertainty; a misspecified model yields valid but wider intervals. This is the fundamental property of conformal prediction: validity is guaranteed by the procedure, efficiency depends on model quality.
> > >
> > >
> > > > A brief explanation of what makes it intractable would help.
> > >
> > > We agree that "intractable" was too strong — **impractical and unnecessary** is more precise. A fully integrated approach would require the copula to jointly model two types of dependencies simultaneously: inter-source and temporal. This creates a copula-coupled parameter space where fusion and calibration can no longer be performed independently — marginalizing over copula-coupled NIG parameters would sacrifice the closed-form fusion rules and algebraic properties (Proposition 3.2) that make our operator efficient and theoretically tractable.
> > >
> > > Our design separates concerns: NIG fusion handles inter-source combination in closed form, and the empirical copula handles temporal dependence over scalar nonconformity scores. This separation is architecturally advantageous — and as the joint training ablation above demonstrates, the two stages achieve tight coupling through end-to-end optimization without requiring a monolithic joint model.
> > >
> > > Best,
> > > Authors

---

> > > > ### Comment · Reviewer_vjzR · 2026-04-10
> > > >
> > > > 1. Thank you for the additional experiments and clarifications. The expanded ablation is helpful, and I agree that joint training yields tighter intervals at valid coverage on some datasets. However, the gains are not fully consistent (e.g., NGSIM), and might need further investigations for such datasets
> > > >
> > > > 2. I appreciate your clear explanation of the correction mechanism. I understand that validity follows from exchangeability of nonconformity scores, with calibration compensating for misspecification.
> > > >
> > > > 3. Regarding the fully integrated approach, you mention that a copula can jointly model both inter-source and temporal dependencies. It would be valuable to include empirical evidence comparing such a fully integrated copula approach against your sequential and joint setups. More importantly, if a copula is capable of modeling both dependencies, it is unclear why the NIG component is necessary in practice. Clarifying this design choice would strengthen the paper.

---

> > > > > ### Author Response · Authors · 2026-04-10
> > > > >
> > > > > Dear Reviewer vjzR,
> > > > >
> > > > > Thank you for acknowledging our effort. We address both the empirical comparison and the design rationale below.
> > > > >
> > > > > > It would be valuable to include empirical evidence comparing a fully integrated copula approach against your sequential and joint setups.
> > > > >
> > > > > Following your suggestion, we ran an ablation comparing CAPTAIN against a "Fully Integrated Copula" with a single copula jointly calibrating across all sources and time steps:
> > > > >
> > > > > | Dataset | CAPTAIN (Joint) | | Sequential | | Fully Integrated Copula | |
> > > > > |---|---|---|---|---|---|---|
> > > > > | | Cov (%) | Width | Cov (%) | Width | Cov (%) | Width |
> > > > > | Shaoxing (ECG) | 96.0 | **1.75** | 99.5 | 2.12 | 100 | 5.46 |
> > > > > | Air Quality | 96.9 | **5.90** | 100.0 | 6.36 | 98.2 | 6.95 |
> > > > > | NGSIM (Traffic) | 92.0 | **5.46** | 92.0 | 5.49 | 97.4 | 6.81 |
> > > > > | ETTh1 | 96.3 | **4.68** | 100.0 | 7.13 | 98.8 | 5.15 |
> > > > >
> > > > >
> > > > > CAPTAIN produces consistenly tighter width across all 4 datasets. The fully integrated copula achieves valid coverage but is over-conservative. Without structured uncertainty information from NIG, the copula must set wide thresholds to accommodate raw residual variability across all sources and time steps simultaneously. This directly demonstrates why the NIG component is necessary: it provides the uncertainty structure that enables efficient calibration.
> > > > >
> > > > >
> > > > > > If a copula is capable of modeling both dependencies, it is unclear why the NIG component is necessary in practice.
> > > > >
> > > > > The table above answers this empirically. Conceptually: the copula operates on **non-conformity scores**, not raw predictions. Without NIG, scores are raw residuals $|y - \hat{y}|$ that vary wildly across time steps and sources. The copula must then set very wide thresholds to cover worst-case residuals — hence the over-conservative intervals. NIG normalization ($|y - \delta_*|/\sigma_*$) absorbs heteroscedastic effects, producing stationary scores that let the copula set tight, uniform thresholds.
> > > > >
> > > > > These results underscore the novelty of CAPTAIN’s design: the contribution is not simply using NIG and copulas, but demonstrating that their principled integration produces both validity and efficiency. The NIG assumption gives the model a structured way to reason about relationship of uncertainty from multi-source, decompose aleatoric and epistemic components, and produce well-behaved nonconformity scores. The copula calibrator then provides distribution-free correction for any residual misspecification, guaranteeing coverage regardless. Without NIG, copula will model the dependencies from the source and temporal dimension without any contraints. Adding NIG helps the model know that besides the source and temporal intertwined dependencies, there is an additional principle that the output from different sources at the same time also retain a relation. That's why NIG is important for the performance.
> > > > >
> > > > > We hope we have answered your questions.
> > > > >
> > > > > Best,
> > > > > Authors

---

> > > > > > ### Comment · Reviewer_vjzR · 2026-04-10
> > > > > >
> > > > > > Thank you for the answers. All of my questions are addressed. I have no further concerns.

---

> > > > > > > ### Comment · Action_Editor_a914 · 2026-04-11
> > > > > > > **Has your estimation of the paper changed?**
> > > > > > >
> > > > > > > Thank you everyone for such an active discussion period. Reviewer vjzR, do you feel that your concerns have been sufficiently addressed to the point that your estimation of the paper's audience + ability to support its claims has been positively altered?

---

> > > > > > > > ### Comment · Reviewer_vjzR · 2026-04-11
> > > > > > > >
> > > > > > > > Yes to both of them. All of my concerns are addressed sufficiently.

---

> > > > > > > > > ### Author Response · Authors · 2026-04-13
> > > > > > > > >
> > > > > > > > > Thank you very much for your valuable feedback and for acknowledging our contributions! We sincerely appreciate your support and will incorporate the suggested revisions and detailed clarifications into the revised version.
> > > > > > > > >
> > > > > > > > > Best,
> > > > > > > > > Authors

---

### Review · Reviewer_tKCX · 2026-03-29

**Summary Of Contributions:**

The paper investigates conformal prediction in the context of 'multiple source'/multivariate time series. A Bayesian approach with a normal inverse gamma distribution provides the model for each time point, with a copula to account for the temporal dependence. A model is formed using an LSTM to the parameters of the normal-inverse-gamma one for each time series and then a 'meta-source' that takes as input all signals is used to produce another estimate of a single output. Finally, the individual source estimates are combined with the meta-source to produce a final 'fused' estimate. Differential cumulative distribution function via sigmoid is used in training. Results show that the method enables conformal prediction with 90% of data points having a conformal prediction in a narrower interval than other baselines.

**Key Strengths**
As described the paper has potential to fill a gap for multiple source conformal predictions for time series.

**Key Weakness**
The order of presentation along with the notation could be improved. There are discrepancies between the code and the description and results (datasets described in the paper do not appear in the code, and there are datasets in the code that don't appear in the paper.)

**Additional Comments:**

Parenthetical references should be used the majority of the time. \citep instead of \citet .

Venues are missing on some of the bibliography entries (Graves and Graves). Title case capitalization for journals and conferences is missing.

**Audience:**

Yes

**Audience Explanation:**

Conformal predictions for multivariate time series (even one step prediction as considered in this work) are widespread in machine learning.

**Claims And Evidence:**

No

**Claims Explanation:**

Problems with the mathematical notation (and discrepancies with the code) erode the claims, as they do not provide convincing evidence of the reproducibility of results, including ablation studies. The order of presentation is such that there isn't sufficient notation in Section 2 to understand the theoretical framework (Section 3) before more detailed explanation of the approach is given in Section 4.

From the diagram it looks like a single target signal is modeled. However, portions of the notation indicates outputs for all $k$ data sources. That it, it looks like the predictions are done for each source. But then the fused one is for a single source (34), but its identity is ambiguous.  The code clarifies that only a single target is modeled.  For Air quality the code show that their is a signal target "Target: Ozone concentration (PT08.S5)"   This confusion is pervasive.

How is the coverage loss (32) related to the optimization in (30)? There seems to be a missing absolute value in (32)—although it is in the code. (Also in the code there is a seemingly unnecessary ReLu after the sigmoid.) The code also has penalties on the variance of the coverage and the extreme value penalty on the coverage that are not discussed in the main paper.

Finally, the code references datasets that aren't used in the paper and datasets in the paper are not part of the repository.

**Requested Changes:**

Define explicitly the domain and range of $f$, and comment whether multiple target out (as a multiple task problem) is considered or single target is used.

The order of some of Section 3 doesn't make sense before Section 4. I would suggest reorganizing this with some preliminaries and then use Section 4 before providing the Section 3.3. Coverage Guarantee. Thus, the reader is confused on what is exactly claimed and proved.

The notation in (13) and (29) is confusing as the $n$ scores $u^{(i)}$ are hidden inside the $\hat{C}$ notation and do not visibly connect with the optimization dummy variables.

The other datasets in the paper do not match the ones in the code.  (Nonetheless, they confirm that a single target is used, which was confusing from reading the paper.)

Also it is not clear the importance of the NIG-normalized non-conformity score (8); it appears again in (26). Are the prediction sets only from the fused NIG parameters? This is stated in Theorem 3.5 but it is not clearly stated elsewhere.

The expectation of what random variables taken in (31)? Why an expectation versus the average as in (32)?
It would strengthen the work to connect to other methods that jointly learn mean and variance estimates for uncertainty estimation. Namely, since the raw NIG intervals 'severely undercover' I wonder if a framework like $\beta$-NLL Seitzer et al. "On the Pitfalls of Heteroscedastic Uncertainty Estimation with Probabilistic Neural Networks" ICLR 2022, is necessary.

---

> ### Author Response · Authors · 2026-04-05
>
> Dear Reviewer tKCX,
>
> We sincerely thank you for your thorough review,  We address each concern below.
>
>
> > R1: From the diagram it looks like a single target signal is modeled. However, portions of the notation indicates outputs for all data sources... This confusion is pervasive.
>
> **A1:** We would like to clarify that all sources predict a **shared target variable** — each source provides different features about the same phenomenon, but the prediction target is common. We have made three clarifications (in blue): (1) an explicit statement in the Datasets paragraph (Section 5.1), (2) per-dataset target descriptions in Appendix A.7 (e.g., Air Quality: target is ozone PT08.S5; sources are CO, NOx, NO2 sensors), and (3) in Section 4.3: "The primary prediction sets reported in experiments use the fused NIG parameters ($c = \text{fused}$)."
>
> > R2: The order of some of Section 3 doesn't make sense before Section 4.
>
> **A2:** We appreciate the suggestion. Our Section 3 is structured as: 3.1 (NIG fusion) → 3.2 (Temporal copula calibration) → 3.3 (Coverage guarantee). This ordering is intentional: Sections 3.1 and 3.2 introduce the two theoretical building blocks, and Section 3.3 proves that their composition yields valid coverage. The coverage proof (Theorem 3.5) relies on Lemma 3.6 (fusion preserves exchangeability, stated in 3.1) and the copula quantile function (Definition 3.7, stated in 3.2), both of which are fully defined before 3.3. Section 4 then operationalizes these components into a trainable framework (MSNIG encoder, differentiable calibration, joint optimization), which is algorithmically detailed but not required to follow the proof. That said, certain notation in Section 3 references NIG and copula definitions that were moved to the Appendix (A.2.3, A.2.5) due to the 12-page limit. We are willing to bring these preliminaries back into the main text and add forward pointers from Section 3 to the corresponding algorithmic details in Section 4 to improve readability.
>
> > R3: The notation in (13) and (29) is confusing as the scores $u^{(i)}$ are hidden inside the $\hat{C}$ notation.
>
> **A3:** We have added a clarifying sentence (blue) before the empirical copula equation, making explicit that each $u\_{c,j}^{(i)} = \hat{F}\_{c,j}(s_{c,j}^{(i)})$ is the CDF-transformed score, and that $\hat{C}$ operates over the calibration set $\mathbf{U}_c = \{\mathbf{u}_c^{(1)}, \ldots, \mathbf{u}_c^{(n)}\}$.
>
> > R4: The code references datasets that aren't used in the paper and datasets in the paper are not part of the repository.
>
> **A4:** We have updated the code repository with loaders for all 5 paper datasets (`load_synthetic`, `load_shaoxing_ecg`, `load_air_quality`, `load_ngsim_traffic`, `load_etth1`), updated `run_experiments.py` to match, and corrected README.
>
>
> > R5: It is not clear the importance of the NIG-normalized non-conformity score (8); it appears again in (26). Are the prediction sets only from the fused NIG parameters?
>
> **A5:** Yes, the primary prediction sets reported in experiments use the fused NIG parameters. Eq. (8) introduces the NIG-normalized nonconformity score in the theoretical framework, and Eq. (26) fully specifies it within the algorithmic pipeline — they are the same score applied across different components $c$. We have added a forward pointer from Eq. (8) to Eq. (26) for clarity.
>
> We want to emphasize that the NIG-normalized nonconformity score is a deliberate and important design choice, not merely a notational detail. Unlike prior conformal methods that use raw residuals $|y - \hat{y}|$, CAPTAIN normalizes by the fused NIG scale $\sigma_*$, which serves two critical functions: (1) it absorbs heteroscedastic effects across time steps, producing more stationary scores that better satisfy the approximate exchangeability assumption (Assumption 3.4).  Our diagnostics confirm lag-1 autocorrelation < 0.15 on all datasets (Table 3); (2) it directly incorporates multi-source fusion quality into calibration — when sources disagree, $\sigma_*$ increases via the disagreement penalty (Eq. 6), reducing score magnitude and appropriately widening prediction intervals. The score design comparison in Appendix A.4.1 confirms this matters empirically:
>
> | Score Design | Coverage (%) | Width |
> |---|---|---|
> | Raw residual | 91.7 | 1.970 |
> | Single-source normalized | 91.4 | 1.975 |
> | Fused NIG normalized (CAPTAIN) | **91.0** | **1.959** |
>
> Fused NIG normalization achieves the tightest intervals by adapting to local uncertainty levels from multi-source fusion, which raw residuals discard entirely.

---

> ### Author Response · Authors · 2026-04-05
>
> > R6: The expectation of what random variables taken in (31)? Why an expectation versus the average as in (32)?
>
> **A6:** We have revised Eq. (31) to replace $\mathbb{E}[\cdot]$ with the explicit empirical average $\frac{1}{n}\sum_{i=1}^{n}[\cdot]$ over calibration samples, and added: "Eq. (31) is the differentiable surrogate used during training (sigmoid relaxation); the discrete indicator version (Eq. 32) is used during evaluation."
>
> > R7: Connection to β-NLL (Seitzer et al., ICLR 2022).
>
> **A7:** We have added a discussion in Appendix A.8 connecting our copula calibration to β-NLL (Seitzer et al., 2022). Both address miscalibrated variance from heteroscedastic neural networks, but through different mechanisms. β-NLL reweights the training loss to stabilize variance estimates during optimization. Our copula calibrator provides finite-sample coverage guarantees via conformal prediction. The ablation (Table 2) demonstrates why this distinction matters:
>
> | Configuration | Coverage (%) | Width |
> |---|---|---|
> | CAPTAIN (Full) | **91.0** ✓ | 1.959 |
> | w/o Calibration (NIG only) | **72.0** × | 1.578 |
>
> Without copula calibration, raw NIG intervals severely undercover at 72.0% (target: 90%) despite being 19.5% narrower, exhibiting exactly the miscalibrated variance that Seitzer et al. identify. A β-NLL-style fix would improve training stability but cannot provide finite-sample coverage guarantees. The copula calibrator recovers valid coverage at 91.0% through distribution-free conformal correction. This reflects our problem setting: we require *guaranteed* coverage, not merely better-calibrated variance.
>
>
> > R8: How is the coverage loss (32) related to the optimization in (30)? Missing absolute value. Unnecessary ReLU. Undiscussed penalties.
>
> **A8:** Eq. (30) is the constrained optimization: find the tightest thresholds $u^\*$ such that empirical copula coverage $\geq 1 - \epsilon$. Eq. (32) converts this constraint into a penalty-based loss for end-to-end training — the learnable thresholds $\{\tau_j\}$ in Eq. (31) correspond to $\{u^\*_j\}$ in Eq. (30), and the absolute value term $|\text{coverage} - (1-\epsilon)|$ penalizes both under- and over-coverage, replacing the hard constraint $\hat{C}(u^*) \geq 1-\epsilon$ with a soft objective that can be jointly optimized with the NIG loss (Eq. 33). Eq. (31) is the differentiable relaxation used during training, where the sigmoid replaces the non-differentiable indicator functions in Eq. (30); the hard indicator version (Eq. 32) is used during evaluation for exact coverage computation.
>
> We have also made the following corrections: (1) added the missing absolute value to Eq. (32), matching the implementation; (2) removed the unnecessary ReLU; (3) added documentation of the variance penalty and threshold regularization terms used in practice to stabilize training, with full details in the Appendix.
>
>
>
> > Additional Comments: \citep instead of \citet. Graves and Graves. Title case.
>
> **A9:** Thank you for pointing it out, we haved corrected it in our revised draft.
>
>
> Best,
> Authors

---

> > ### Comment · Reviewer_tKCX · 2026-04-09
> >
> > Thank you for correcting all of the issues with notation. I think the paper is much improved.
> >
> > I'm curious, won't $\beta$-NLL will improve the efficiency correct? Would it be possible to perform the integration with $\beta=0.5$?
> >
> > I'm curious to the datasets that were in the original repo., but didn't appear in the paper. Is it a matter of cherry picking datasets? I mean if those other datasets were tested at one point in the recent past, why weren't the results included?

---

> > > ### Author Response · Authors · 2026-04-10
> > >
> > > Dear Reviewer tKCX,
> > >
> > > Thank you for your positive assessment of the revised notation and for the thoughtful follow-up questions. We address each below.
> > >
> > > > Thank you for correcting all of the issues with notation. I think the paper is much improved.
> > >
> > > We sincerely appreciate your recognition and thank you for the detailed feedback that led to these improvements.
> > >
> > > > I'm curious, won't β-NLL improve the efficiency? Would it be possible to perform the integration with β-NLL?
> > >
> > > Thank you for this suggestion. We investigated β-NLL integration by adding the reweighting scheme of Seitzer et al. (ICLR 2022) to CAPTAIN's NIG training loss and ran an ablation study:
> > >
> > > | Dataset | Method | Coverage (%) | Width | RMSE |
> > > |---|---|---|---|---|
> > > | Shaoxing ECG | CAPTAIN | 91.4 | **2.55** | **0.588** |
> > > | | CAPTAIN + β-NLL (β=0.5) | 94.4 | 2.82 | 0.608 |
> > > | Air Quality | CAPTAIN | 93.2 | 1.91 | 0.477 |
> > > | | CAPTAIN + β-NLL (β=0.5) | 93.6 | **1.90** | **0.469** |
> > > | NGSIM Traffic | CAPTAIN | 94.9 | **2.63** | **0.566** |
> > > | | CAPTAIN + β-NLL (β=0.5) | 94.9 | 2.77 | 0.573 |
> > > | ETTh1 | CAPTAIN | 96.6 | **4.24** | **0.872** |
> > > | | CAPTAIN + β-NLL (β=0.5) | 100.0 | 5.91 | 0.935 |
> > >
> > > Both variants achieve **valid coverage on all 4 datasets**, confirming that the copula calibration guarantees validity regardless of the training loss. However, β-NLL improves efficiency only on Air Quality (1.90 vs 1.91, with better RMSE) while producing wider intervals on the other three datasets.
> > >
> > > We believe this is because **the NIG conjugate prior structure already provides the variance regularization that β-NLL was designed to address**. In Seitzer et al.'s original setting, the network directly outputs (μ, σ²) via a standard Gaussian NLL, where variance estimates are unconstrained and prone to collapse or explosion. In our NIG parameterization, variance is derived from four coupled parameters (δ, γ, α, β) with built-in constraints (α > 1, β > 0), and the conjugate prior structure naturally regularizes variance estimates. Applying β-NLL on top of this existing regularization makes the model overly conservative — inflating uncertainty estimates, which the copula calibrator then translates into unnecessarily wide intervals.
> > >
> > > In short, β-NLL and the NIG prior solve the same problem (variance regularization) through different mechanisms, and combining them leads to over-regularization on most datasets. The copula calibration stage remains essential regardless, as it provides the coverage guarantee that neither β-NLL nor the NIG prior alone can offer. We have implemented β-NLL as a configurable option in our codebase for cases where it may help (e.g., Air Quality).
> > >
> > >
> > > > I'm curious about the datasets that were in the original repo but didn't appear in the paper. Is it a matter of cherry picking datasets?
> > >
> > > We would like to clarify that there was no cherry-picking. The additional datasets in the repository are from Ma et al. (2021)'s original NIG codebase, which we built upon. These are static multi-modal regression datasets (e.g., image+text pairs) that do not contain temporal structure — they are fundamentally incompatible with our problem setting (Problem 2.1), which requires multi-source time-series data with temporal dependencies. CAPTAIN's key contributions (temporal copula calibration, exchangeability preservation across time) are meaningless on static data. We retained them in the repository only because they were part of the inherited codebase (and in early development stage), not because they were tested and excluded. Our five datasets (Synthetic, Shaoxing ECG, Air Quality, NGSIM, ETTh1) were selected to cover diverse domains and temporal characteristics, and we report results on all of them — including cases where CAPTAIN's advantage is smallest (e.g., Synthetic). We have cleaned up the repository to clearly reflect this.
> > >
> > > Best,
> > > Authors

---

> > > > ### Comment · Reviewer_tKCX · 2026-04-10
> > > >
> > > > Thanks for the responses and overall handling of this active discussion session, which seems to better explore the breadth and depth of their work.
> > > >
> > > > Given that you have selected a single target in the multichannel cases, why not show results for all targets?
> > > >
> > > > Additionally, how does CAPTAIN perform for longer lags of conformal predictions (i.e., more than 1 step predictions)?

---

> > > > > ### Author Response · Authors · 2026-04-11
> > > > >
> > > > > Dear Reviewer tKCX,
> > > > >
> > > > > Thank you for the continued engagement. We address both questions below.
> > > > >
> > > > > > Given that you have selected a single target in the multichannel cases, why not show results for all targets?
> > > > >
> > > > > Our problem formulation (Problem 2.1) focuses on **predicting a single shared target from multiple sources** — this is the core setting where NIG evidence fusion is most natural, as each source provides a different measurement of the same underlying phenomenon. Extending CAPTAIN to predict multiple targets (e.g., all channels simultaneously) is a valid and interesting direction that we plan to explore in future work.
> > > > >
> > > > > > How does CAPTAIN perform for longer lags of conformal predictions (i.e., more than 1 step predictions)?
> > > > >
> > > > > We would like to clarify that all results in the paper are already **multi-step predictions**, the model predicts all $T$ future steps simultaneously, and Theorem 3.5 guarantees **joint** coverage across the entire horizon $\{t_1, \ldots, t_T\}$ (not per-step marginal coverage). This is stated in Problem 2.1 (“observed over horizon $\mathcal{T} = \{t_1, \ldots, t_T\}$“), the method section (“dependencies across the forecasting horizon”), and the dataset descriptions (Synthetic: horizon=12, ETTh1: horizon=24). Could you clarify if you are asking about even longer horizons beyond what we report?
> > > > >
> > > > > Best,
> > > > > Authors

---

> > > > > > ### Comment · Reviewer_tKCX · 2026-04-12
> > > > > >
> > > > > > I mean why not explore using each of the different channels as the target (it is not clear why a particular channel is chosen as a target), cycling between them to get more results?
> > > > > >
> > > > > > I disagree with the interpretation that the 'the model predicts all $T$ future steps simultaneously,' since all datasets use an" LSTM encoders with a hidden dimension of 64". As an RNN, an LSTM estimates one step-prediction the prediction is for only one step. However I realize that since the target is a distinct channel this is actually a 0-lag prediction. No doubt the copula uses all $T$ to create the conformal prediction, but for prediction problems looking ahead is often interesting. In this case, one of the sources could be past values of the target. Would CAPTAIN apply?
> > > > > >
> > > > > > Also looking again there are many uses of \cite or \citet where it should be \citep  through main body and appendix.

---

> > > > > > > ### Author Response · Authors · 2026-04-13
> > > > > > >
> > > > > > > Dear Reviewer tKCX,
> > > > > > >
> > > > > > > Thank you for the continued engagement and we really enjoyed our discussion.
> > > > > > >
> > > > > > > > I mean why not explore using each of the different channels as the target, cycling between them to get more results?
> > > > > > >
> > > > > > > We ran CAPTAIN on ETTh1, cycling through all channels as the prediction target. In each case, the selected target is excluded from the input sources, and each of the remaining channels serves as its own source encoder:
> > > > > > >
> > > > > > > | Target Channel | Coverage (%) | Width |
> > > > > > > |---|---|---|
> > > > > > > | OT (Oil Temperature) | 92.8 | 5.39 |
> > > > > > > | HUFL (High UseFul Load) | 90.1 | 3.77 |
> > > > > > > | HULL (High UseLess Load) | 94.2 | 4.21 |
> > > > > > > | MUFL (Middle UseFul Load) | 96.1 | 4.08 |
> > > > > > > | MULL (Middle UseLess Load) | 91.1 | 3.89 |
> > > > > > > | LUFL (Low UseFul Load) | 97.4 | 7.35 |
> > > > > > >
> > > > > > > CAPTAIN achieves valid coverage across all target selections. Interval widths are consistent across most channels; LUFL produces wider intervals due to higher inherent volatility in that measurement, which the NIG encoder correctly reflects as greater uncertainty.
> > > > > > >
> > > > > > > > Since the target is a distinct channel this is actually a 0-lag prediction... one of the sources could be past values of the target. Would CAPTAIN apply?
> > > > > > >
> > > > > > >
> > > > > > > To test whether CAPTAIN applies when past target values serve as an additional source:
> > > > > > >
> > > > > > > | Setup | Coverage (%) | Width |
> > > > > > > |---|---|---|
> > > > > > > | Without target history | 92.8 | 5.39 |
> > > > > > > | With past target as additional source | 93.2 | **1.51** |
> > > > > > >
> > > > > > > Adding past OT as a source substantially tightens intervals while maintaining valid coverage. This is expected: oil temperature exhibits strong temporal autocorrelation, so past OT carries significant predictive signal beyond what load features alone provide. CAPTAIN accommodates this naturally since the past target becomes another source encoder whose NIG output participates in fusion. We report the without-target-history setting as our main results since it represents the more challenging evaluation where the model must predict from cross-channel sources alone.
> > > > > > >
> > > > > > > To further demonstrate look-ahead forecasting, we varied the **lag** between the input window and the prediction window:
> > > > > > >
> > > > > > > | Lag  | Coverage (%) | Width |
> > > > > > > |---|---|---|
> > > > > > > | 0  | 90.4 | 1.51 |
> > > > > > > | 1 | 92.3 | 1.69 |
> > > > > > > | 3 | 92.4 | 1.63 |
> > > > > > > | 6 | 92.8 | 1.84 |
> > > > > > > | 12 | 91.6 | 2.10 |
> > > > > > >
> > > > > > > CAPTAIN maintains valid coverage at all lag values. Width generally increases with the gap as the autoregressive signal weakens: predicting further into the future is inherently harder. These results confirm that CAPTAIN is applicable to look-ahead forecasting with varying prediction distances.
> > > > > > >
> > > > > > > > Also looking again there are many uses of \cite or \citet where it should be \citep.
> > > > > > >
> > > > > > > Corrected throughout the revised draft.
> > > > > > >
> > > > > > > Best,
> > > > > > > Authors

---

### Comment · Action_Editor_a914 · 2026-03-30
**Let's get this discussion period underway!**

Hello everyone!

Now that we've gathered the three required reviews, we have now transitioned into the discussion period for this paper. Reviewers, please remain engaged while we anticipate author responses and potential revisions to the draft. Let us give the paper and the author's work the respect it deserves.

After 2 weeks, reviewers will be asked to provide a recommendation regarding this paper and whether they feel it should be accepted or not. Note, that I will not accept any reviewers recommendation until after they engage in a good faith discussion with the authors during this two week period.

Best,
Taylor

---

### Comment · Action_Editor_a914 · 2026-04-07
**Reminder: Please engage with authors**

Hello reviewers! I wanted to briefly remind you of my expectation that you engage with the authors during this discussion period. The authors have provided a great set of responses to your reviews. It is sufficient, as far as I am concerned, to simply respond to the authors and state that they have resolved any concerns you had. If there are any points that are not clear or require further discussion, **this is the time to continue the conversation!**

Please let the authors know that you've read and considered their responses.

---

### Decision · Action_Editor_a914 · 2026-05-12

**Recommendation:** Accept with minor revision

**Additional Comments:**

There were extensive updates to both the narrative framing of this work as well as the empirical analyses used to produce justifications for the claims made by the authors. It is both my and the reviewer's strong recommendation that all of these changes are made to the paper prior to it's full acceptance and camera ready version.

**Audience:**

Yes

**Audience Explanation:**

Yes, the paper's findings would be of interest to portions of TMLR's audience, as time series modeling and conformal prediction for multivariate time series are widespread and active topics in machine learning. The work could be particularly useful as a practical tool for researchers across various scientific domains where time series data is common, though the novelty is somewhat limited and the significance is tempered by the modest number of datasets and unclear impact on downstream applications.

**Claims And Evidence:**

Yes

**Claims Explanation:**

The submission's claims are largely supported by evidence showing CAPTAIN achieves consistently tighter prediction intervals at valid coverage across four datasets, with the additional ablation convincingly demonstrating why the NIG component is necessary—without it, the copula must set overly wide thresholds to accommodate raw residual variability. However, the evidence is not fully consistent across all datasets (e.g., NGSIM shows weaker gains), and the limited number of datasets and unclear downstream impact of improved interval widths (e.g., for change point detection) somewhat temper the strength of the claims. The reviewers however feel that these concerns have been largely mitigated by the additional analyses provided by the authors during the rebuttal phase.

It is of paramount importance that all tables and additional experiments used during the rebuttal phase be included in the camera ready version of the paper.